# A mouse tissue transcription factor atlas

Quan Zhou[1,*], Mingwei Liu[1,*], Xia Xia[1,*], Tongqing Gong[1], Jinwen Feng[1,2], Wanlin Liu[1], Yang Liu[3], Bei Zhen[1], Yi Wang[4], Chen Ding[1,3] & Jun Qin[1,3,4]

Transcription factors (TFs) drive various biological processes ranging from embryonic development to carcinogenesis. Here, we employ a recently developed concatenated tandem array of consensus TF response elements (catTFRE) approach to profile the activated TFs in 24 adult and 8 fetal mouse tissues on proteome scale. A total of 941 TFs are quantitatively identified, representing over 60% of the TFs in the mouse genome. Using an integrated omics approach, we present a TF network in the major organs of the mouse, allowing data mining and generating knowledge to elucidate the roles of TFs in various biological processes, including tissue type maintenance and determining the general features of a physiological system. This study provides a landscape of TFs in mouse tissues that can be used to elucidate transcriptional regulatory specificity and programming and as a baseline that may facilitate understanding diseases that are regulated by TFs.

[1] State Key Laboratory of Proteomics, Beijing Proteome Research Center, Beijing Institute of Radiation Medicine; National Center for Protein Sciences (The PHOENIX Center, Beijing), Beijing 102206, China. [2] Institute of Biomedical Sciences and School of Life Sciences, East China Normal University, Shanghai 200241, China. [3] State Key Laboratory of Genetic Engineering and Collaborative Innovation Center for Genetics and Development, School of Life Sciences, Institutes of Biomedical Sciences, Fudan University, Shanghai 200032, China. [4] Alkek Center for Molecular Discovery, Verna and Marrs McLean Department of Biochemistry and Molecular Biology, Department of Molecular and Cellular Biology, Baylor College of Medicine, Houston, Texas 77030, USA. * These authors contributed equally to this work. Correspondence and requests for materials should be addressed to C.D. (email: chend@fudan.edu.cn) or to J.Q. (email: jqin@bcm.edu).

Approximately 1,500 transcription factors (TFs) are encoded in the mammalian genome[1] and constitute the second largest gene family, with the immunoglobulin superfamily being the largest. TFs can be grouped into different families, depending on the structure of their DNA-binding domains (DBDs), and each family prefers to bind a specific DNA consensus sequence. Specific recognition of a response element (RE) is essential for gene regulation in response to developmental cues and environmental signals. These DNA sequences are recognized by TFs that recruit co-regulators and the transcriptional machinery[2]. The binding of TFs to DNA determines the strength and resident time of gene regulation[3].

TFs regulate almost every aspect of life[1,4,5]. Tissue specificity is enabled by temporal and spatial gene expression patterns, which are, in turn, driven by TFs[6,7]. This process involves important roles, such as DNA-binding TFs interacting with the cis-elements, including promoters, enhancers and silencers, of the genes they activate or repress[8]. TFs are driving forces in tissue development and tissue identity maintenance. For example, Hnf4a and Hnf1a are critical for liver function[9] and Nkx2-1 expression is related to the peripheral airways and small bronchioles in the lung[10]. These studies have demonstrated the close relationship between TFs and tissue biological processes. Therefore, surveys of TF expression patterns and DNA-binding activities in animals could advance our understanding of how tissue specificity is determined and maintained.

Previously, a variety of genome/transcriptome technologies have been employed to investigate mammalian TFs at high resolution and depth. Many studies have inferred TF expression through mRNA expression profiling using RNA sequencing (RNA-seq) combined with genome promoter analysis[11]. Chromatin immunoprecipitation (ChIP) coupled with microarrays or sequencing is another revolutionary strategy that enables the genome-wide analysis of TF binding[12]. Other methods that study TFs and gene regulation networks, such as yeast one-hybrid assays[13,14] and genome-wide DNase footprints[15] are also widely used to survey TFs interaction with DNA. These technologies have been utilized in the Encyclopedia of DNA Elements (ENCODE)[16] and Functional Annotation of Mammalian (FANTOM)[17] genome projects.

While these studies have led to the construction of models of TF actions, there are critical issues that remain unresolved. First, the correlation coefficient between mRNA and protein abundance is low and is insufficient to predict protein expression levels from quantitative mRNA data[18,19]. Second, ChIP-seq measures the binding sites for one TF at a time and can only obtain data for limited numbers of TFs because of constraints by reagents and experimental throughput. To date, screening TFs and subsequently illuminating their activities at proteome scale remains challenging. Thus, one clear and immediate task is to map TFs at the protein level and determine their DNA-binding activities in different organs/tissues[12].

Towards this goal, we recently developed an approach that permits identification and evaluation of the DNA-binding activity of endogenous TFs at the proteome scale. Using a synthetic DNA containing a concatenated tandem array of the consensus TFREs (catTFRE) as an affinity reagent, the TF sub-proteome can be identified to the TF sub-transcriptome level in cell lines and tissues using mRNA-seq[20].

In this work, we generate a quantitative proteome data set of over 900 TFs in mouse tissues. By integrating multiple-omics data, we present a TF network of major organs of the mouse, allowing data mining and generating knowledge to understand the roles of TFs in various biological processes. This resource can be used to elucidate transcriptional regulation and may help understand diseases that are regulated by TFs.

## Results

**Deep TF DNA-binding activity profiling of mouse tissues.** We profiled the TF DNA-binding activities from 32 histologically normal mouse tissues, including 24 adult tissues and 8 fetal or reproduction-related tissues, to obtain a panoramic view of TF activity in mouse tissues (Fig. 1a). Endogenous TFs were enriched by the catTFRE approach[20] and fractionated by sodium dodecyl sulfate (SDS)–polyacrylamide gel electrophoresis. The tryptic peptides were analysed on high-resolution Orbitrap MS instruments (Orbitrap Q-Exactive) (Fig. 1b, Supplementary Fig. 1a,b). We measured 3–7 biological replicates from the 24 adult tissues until we obtained at least three measurements that showed a correlation coefficient of $R > 0.8$ (Supplementary Fig. 1c). One or two biological repeats for the fetal or reproduction-related tissues were performed (Supplementary Fig. 1d). A website 'Mouse TF Atlas' was developed to host the data (www.tfatlas.org) (Fig. 1c).

Aided by annotation in the DBD database (http://dbd.mrc-lmb.cam.ac.uk/DBD), we could assign 941 of the identified proteins as TFs, representing 60% of the genes encoding TFs. In the adult and fetal tissues, 907 and 587 TFs were identified, respectively; among them, 354 and 34 TFs were unique to the adult tissues and fetal tissues, respectively (Supplementary Fig. 1e and Supplementary Data 1).

All DNA-binding transcription factor (DBTF) families annotated in the DBD database were recovered in our data set. Notably, components of the TF families belonging to AT hook, TDP, RFX, IRF, STAT, SAND, CP2 and AP2 were all detected, and the ZNF-GATA, NR, IPT/TIG, DM and MADs-box families achieved deep coverage, with only one or two components missing. In contrast, approximately 40% and 78% of the ZNF-C2H2 and homeodomain family members, respectively, were detected (Fig. 2a; Supplementary Data 1).

The number of TFs identified in each tissue varied, ranging from 173 in skeletal muscle to 448 in thymus (Fig. 2b). We used FOT (fraction of total), the portion of TF expression in all detected TFs in a particular tissue, as an indicator of TF abundance. We found that the most abundant TF in each tissue accounted for at least 10% of the total TF abundance and that the 4 most abundant TFs accounted for $>30\%$ of the TF abundance in the corresponding tissue. This phenomenon was most profound in intestine, spinal cord, seminal vesicle and lung; for instance, Hmgb2 represented over 60% of the total TF abundance in small intestine, and Tfam accounted more than half of the total TF abundance of spinal cord, seminal vesicle and lung (Fig. 2c and Supplementary Data 1).

Geiger and colleagues reported a proteome profiling of 28 mouse tissues[21]. When compared with the proteome profiling data set, the catTFRE showed a clear advantage in enriching endogenous TFs with the identification of 941 compared to 151 identified by proteome profiling (Supplementary Fig. 1e). Correlation analysis of the overlapping TFs in the 13 tissues in both data sets showed that the Spearman's rank correlation coefficient ranged from 0.046 (liver) to 0.401 (spleen), revealing a difference between TF expression levels and their DNA-binding activities.

We mined hundreds of published literatures (Supplementary Data 1) to construct a library for well-characterized TFs in the 13 overlapped tissues. As shown in Supplementary Data 1, the catTFRE successfully detected most of these TFs in tissues (76 out of 85), while profiling only identified few of them (6 out of 85).

Transcriptional co-regulators (TCs) play critical roles in transcriptional regulation by interacting and cooperating with the TFs. The catTFRE data set also contained 523 TCs (Supplementary Data 1). An L-shaped distribution pattern was observed among the 32 tissues (Supplementary Fig. 2a).

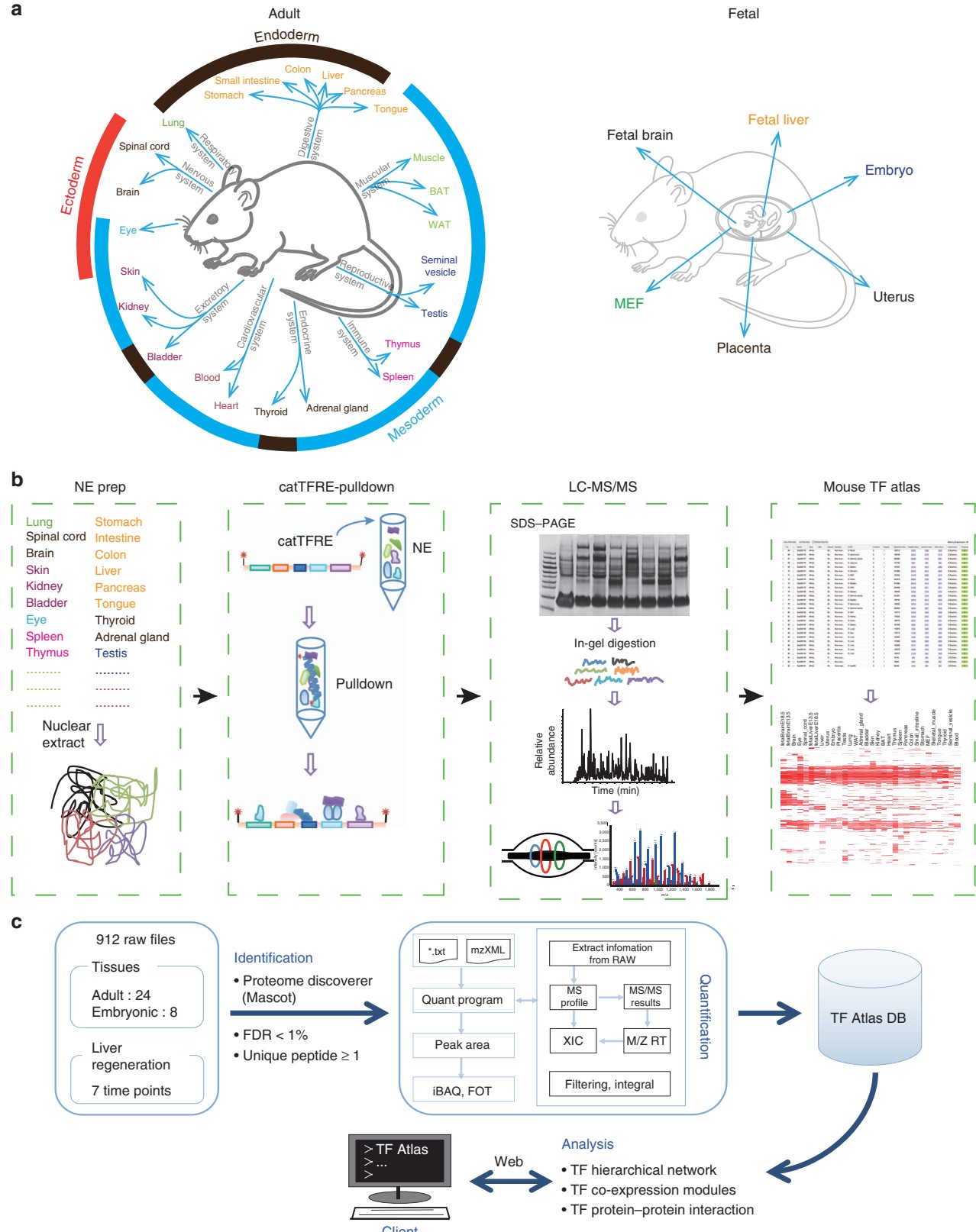

**Figure 1 | Overview of the mouse tissue TF atlas workflow.** (**a**) The 24 adult mouse tissues and 8 fetal tissues analysed to generate a draft map of the mouse TF atlas are shown. The same font colour indicates that the tissues are from the same physiological system. The ambient arcs show the layers from which the tissues developed. Red: ectoderm; Blue: mesoderm; Brown: endoderm. (**b**) Nuclear extract (NE) were prepared from each tissue, and then the catTFRE pull down was carried out and analysed by LC-MS/MS to obtain the primary data. (See the Methods section for details). (**c**) The data were analysed with a homemade platform and stored in the mouse TF atlas database. (See the Methods section for details).

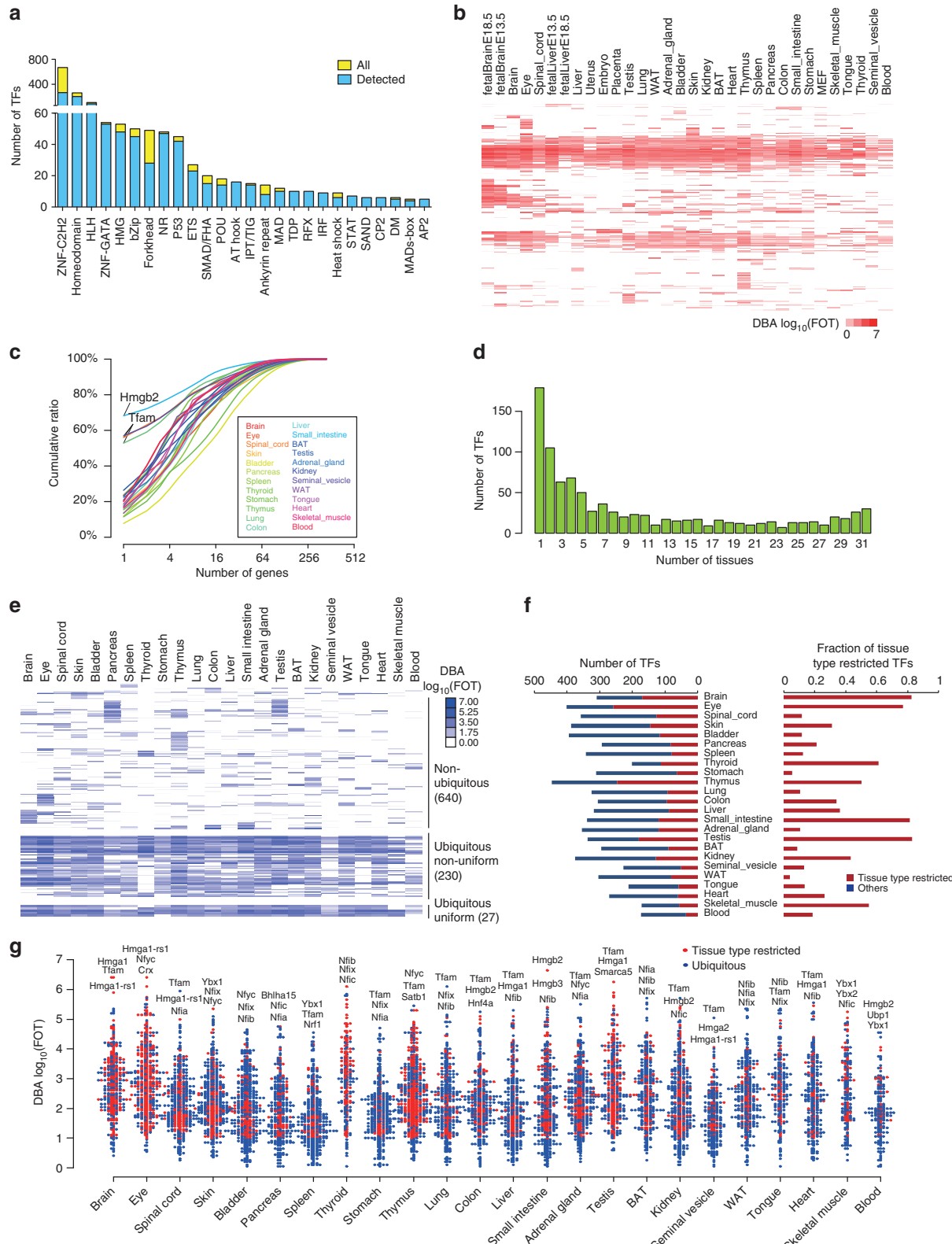

**Figure 2 | TF identifications in 24 adult and 8 fetal mouse tissues.** (**a**) Number of TFs identified in each TF family. The TFs were classified into families according to their DNA-binding domains. TFs with multiple DBDs were classified in each of their respective families. The number of TFs in each family that was detected using the catTFRE strategy is plotted. (**b**) Heatmap of TF DNA-binding activity in 24 adult mouse tissues and 8 fetal tissues; the TFs are shown in the rows, and the tissues are shown in the columns. DBA: DNA-binding activity. (**c**) Cumulative protein abundance from the highest to lowest in the 24 adult tissues. (**d**) Number of tissues in which the TFs are expressed. (**e**) Heatmap for non-ubiquitous (71.3%), ubiquitous-uniform (3.1%) or ubiquitous non-uniform (25.6%) TF. The colour bar on the right indicates the relative expression abundance. (**f**) The number of tissue restricted TFs (ttrTFs) and their fractions in total TFs in different tissues. (**g**) Distributions of ubiquitous and ttrTFs and their relative abundances in 24 adult tissues. The abundance of the TFs spans almost seven orders of magnitude. The names of the top three most abundant TFs are listed.

Interestingly, TCs in general showed lower tissue-specificity scores (TSPS)[22], the relative entropy to evaluate the distance between the observed TF expression pattern and the uniform expression pattern across all tissues, than TFs (P value = 3.86E-16), suggesting their more uniform distribution and essential functions in the whole body (Supplementary Fig. 2b,c).

**Ubiquitous and tissue-restricted DBTFs in mouse tissues**. To investigate the TF expression patterns in different tissues, we plotted the number of TFs that were detected in different tissues and found that less than 32% of the total number of TFs were detected in more than 12 tissues (half of 24 adult tissues) (Fig. 2d).We calculated the median and maximum FOT values in all tissues where they were detected and used them to perform a density plot analysis of the TF expression patterns across the 24 adult tissues. This analysis stratified the TFs into three categories: non-ubiquitous TFs (non-ubiTFs, 640 TFs, accounting for 71.3% of the identified TFs), which were highly expressed in only a few tissues with a transformed median expression value of < 0.5; ubiquitous TFs (ubiTFs, 257 TFs, 28.7%), which were expressed in a wide variety of tissues with a transformed median expression value of > 0.5. Among them, the expression of 27 TFs exhibited a maximum value of less than ten times the median value, indicating a ubiquitous-uniform distribution (27 TFs, 3.1%); the rest of the TFs can be classified as ubiquitous-non-uniform (230 TFs, 25.6%), with a maximum expression value exceeding ten times the medium value (Fig. 2e, Supplementary Fig. 2d and Supplementary Data 2). Consistent with their wide distribution, the ubiquitous-non-uniform TFs mainly function in generic biological processes, such as circadian rhythms, cell cycle, cell growth and chromatin remodelling.

We define TFs that are expressed in a tissue at levels that are at least ten times higher than the median value of all adult tissues as tissue type restricted TFs (ttrTFs). The extreme case is the tissue-specific TF, which is expressed only in one tissue. A large number of ttrTFs were identified in the central nervous system, the immune system (thymus), and the reproductive system (testis), whereas a smaller number of ttrTFs were identified in the metabolic system (intestine, liver, stomach and adipocytes) (Fig. 2f,g). The TF specificity in adult tissues was further validated by qPCR (Supplementary Fig. 2e and Supplementary Data 2). Importantly, TF specificity at the mRNA levels is largely consistent with their DNA-binding activities.

**Transcription network of the NRs and other TF families**. Studies on the mRNA expression profile of TFs offered a simple and powerful way to obtain highly relational information regarding the physiological functions of the individual proteins and the protein families. One of the most successful examples is the anatomical profiling of NRs by the Nuclear Receptor Signaling Atlas (NURSA) organization[23–25]. The mRNA profiling of NRs defined a ring of NR physiology, dividing the NR regulatory network along two physiological paradigms: (1) reproduction, development and growth; and (2) nutrient uptake, metabolism and excretion. These studies reveal a transcriptional circuitry that extends beyond individual tissues to form a mega network governing physiology on an organismal scale.

Here, we were able to reliably detect 47 of the 49 NRs from 32 mouse tissues, with the exception of Nr1h5 and ESR2 (Fig. 3a). Among them, 7 NRs were expressed in all tissues (not including blood), 10 NRs were expressed in more than half of but not all tissues and 30 NRs were restricted to less than 50% of the tissues (Fig. 3b, Supplementary Fig. 3a and Supplementary Data 3).

While Nr1h5 mRNA expression was detected at very low levels in three tissues[23], TFRE-bound Nr1h5 was detected in one experiment in mouse liver TFRE experiment and ESR2 in one mouse embryonic fibroblast (MEF) TFRE experiment with transfected over-expression of Hnf1a, Pdx1 and Bhlha15.

Unsupervised hierarchical clustering of the NR DNA-binding activity revealed two major clusters, which can be further divided into five sub-clusters (Fig. 3a). Cluster IC mainly includes (sex)-steroid hormone receptors, such as AR, ER, PR and DAX-1. Cluster IB includes NRs that are ubiquitously expressed in adult tissues and are expressed at low levels in embryos. Cluster IIB NRs are predominantly expressed in the digestive system, whereas the NRs in Cluster IB have a more widespread expression in different tissues.

In general, our clusters are consistent with the previous one derived from the mRNA profiling[23] but show some differences (Fig. 3a). Esrra/ERRa, Nr2c1/TR2, Nr2f6/EAR2 and Nr6a1/GCNF, which were once classified in Cluster lipid metabolism and energy hemostasis, are now classified in Cluster I as ubiquitous NRs and reproduction; Nr1h2/LXRb and Nr3c2/MR that were once classified in Cluster central nervous system and circadian function are now classified in Clusters IIA and IIB in the digestive system. The current classification is consistent with their functions that were uncovered in more recent studies. For example, Nr1h2 regulates genes involved in liver metabolism and cholesterol uptake[26,27], and Nr6a1 plays an important role in germ cell development during gametogenesis[28]. Nr2c1 and Nr2c2 form the direct repeat erythroid definitive complex, which plays a fundamental role in early embryogenesis and embryonic stem cell proliferation[29]. Notably, these two NRs were classified to the same cluster but were divided into two groups based on the transcriptome data. A modified version of 'The Nuclear Receptor Ring of Physiology'[24] is illustrated in Fig. 3c. A minor difference in classification between current and previously proposed one revealed the diverse and complementary information provided by TF DNA-binding activity at protein level and gene expression at mRNA level.

Other TF families can be similarly analysed to uncover the 'Ring of Physiology'. For instance, a cluster of the Fox family (Foxa1, Foxa2, Foxa3 and Foxf1) was enriched in the digestive system; another cluster of the Hmg family (Sox4, Sox14, Hmga2, Sox2, Sox1 and Hmgn3) was predominantly expressed in the nervous system (Fig. 3d and Supplementary Fig. 3b). The propensities of TF families in different tissues suggested a connection between TF DNA-binding domains and biological functions (Supplementary Data 3).

**Illuminating the dark proteome of TFs**. We investigated the co-expression of TFs in 24 adult tissues using TFs that were expressed in more than four tissues. Using Pearson correlation coefficient of 0.4 as a cutoff to define a positive correlation between TF pairs[30], we obtained 37 tightly related TF modules that showed excellent co-expression patterns (Fig. 4a and Supplementary Data 4).We summed the average intensity of each module and then performed hierarchical clustering to reveal the relationship among the 37 TF modules. These modules segregate into six distinct clusters (Fig. 4b). Their Gene Ontology (GO)[31] term enrichments revealed their potential functions (Fig. 4c) and connections with tissue distribution patterns. For instance, many TFs in the Cluster I predominantly function in oxidation-reduction and vitamin/steroid/lipid metabolism and are highly enriched in liver, colon and small intestine. Functions of unknown proteins may be inferred by the known functions of other members of the module, thereby shedding light on the 'dark proteome'. Twenty-two out of the 37 modules appeared to

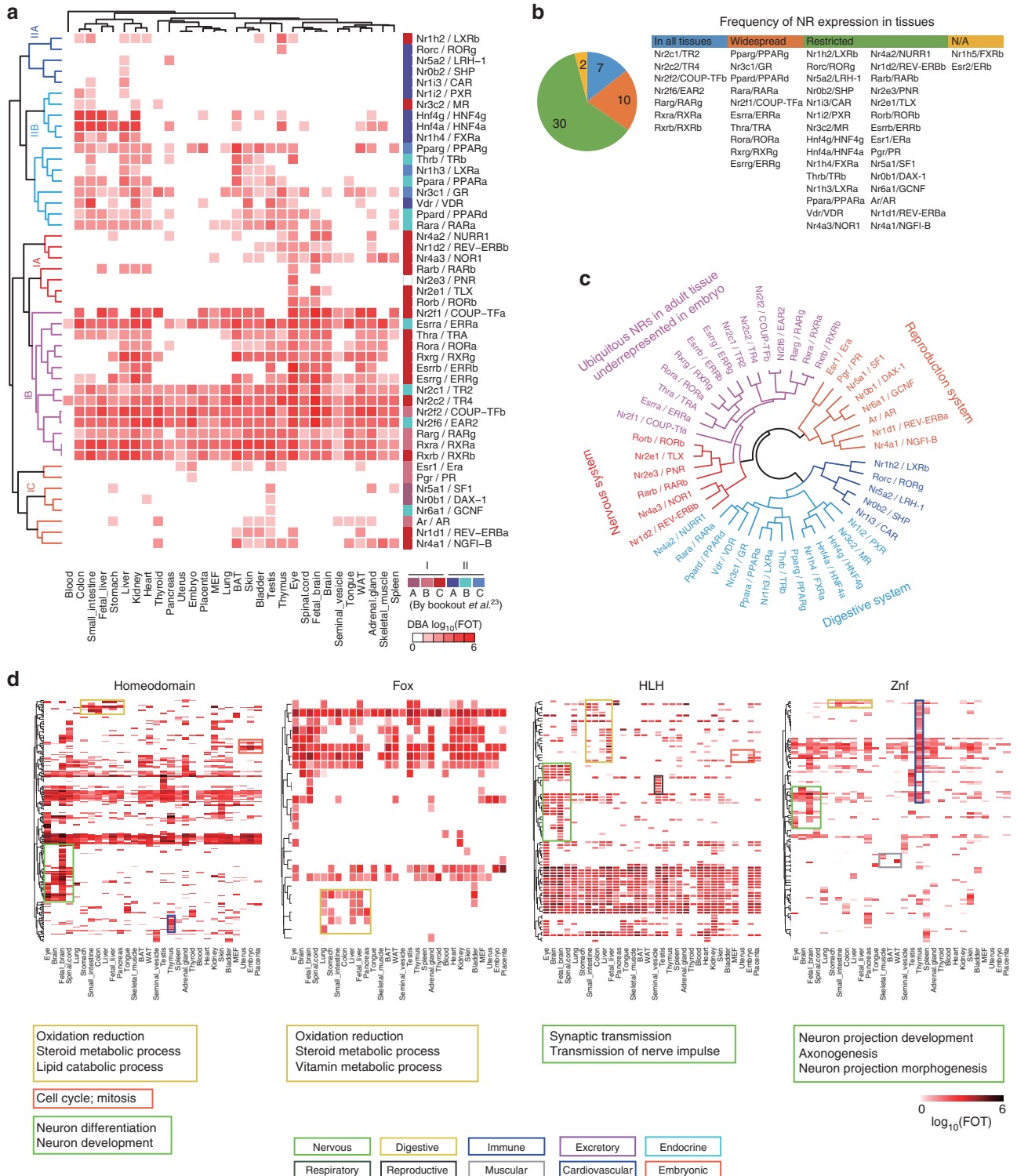

**Figure 3 | Global profiling of TF families in mouse tissues.** (**a**) Unsupervised hierarchical clustering of DNA-binding activities of nuclear receptors (NRs) and the comparison with mRNA expressions. NRs were grouped into two main clusters (I and II) and five sub-clusters (IA, IB, IC, IIA and IIB). The comparison with mRNA is shown on the right colour block. (**b**) Statistics of NR expressions in different tissues. The number of NRs expressed in various tissues is indicated in the pie chart. Seven NRs were expressed in all tissues, ten NRs were expressed in more than half of but not all tissues and were labelled 'widespread', and 30 NRs were restricted to less than 50% of tissues and were named 'restricted'. (**c**) The 'NR Ring of Physiology' derived from DNA-binding activity. The dendrogram is depicted based on the hierarchical clustering, revealing NR clusters in different tissues. (**d**) Unsupervised hierarchical clustering of other TF families. Blocks with different colours represent different physiological systems. The annotations in the blocks are the GO enrichment terms for the TGs that were co-regulated by at least two TFs in the block.

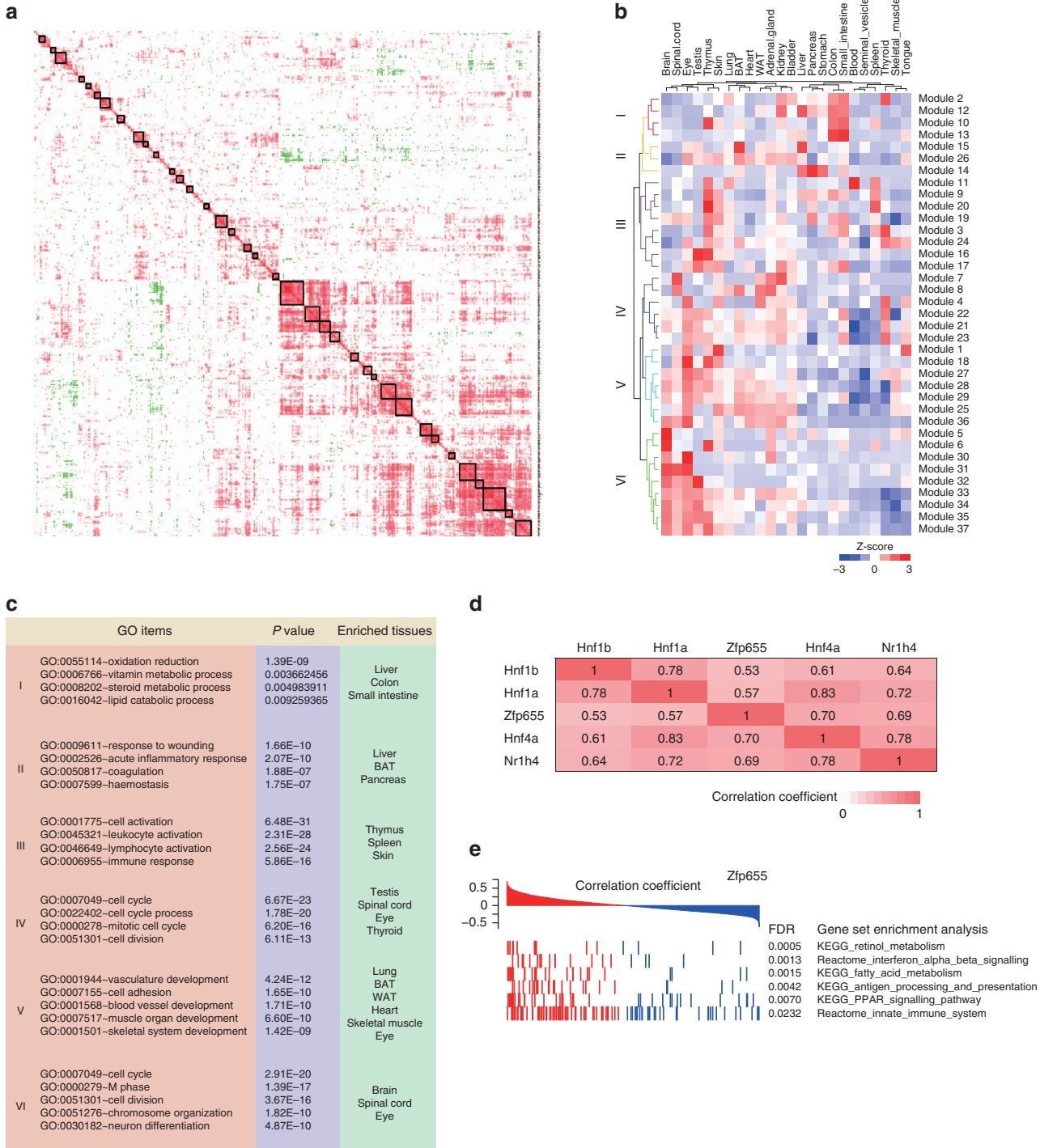

**Figure 4 | The DBTF co-expression network. (a)** Co-expression blocks of TFs. Thirty-seven co-expression blocks of TFs were identified in 24 adult organs. Pearson's correlation coefficient of 0.4 was used as a cutoff, and Kendall's Tau test was applied to test for significance. **(b)** Unsupervised hierarchical clustering of the 37 co-expression modules. Six clusters were grouped according to their different tissue distribution patterns. **(c)** GO term enrichments and tissue distributions of the six clusters derived from the 37 co-expression modules. **(d)** Details of Module #12. The correlation coefficients between the TFs in Module #12 are listed in the table. **(e)** GSEA terms of Zfp655 suggest Zfp655 functions in metabolism and immunity.

have functional enrichment, and predicted functions of 131 of the 156 TFs matched their functions revealed by the previous publications. For example, the Module #1 contains six TFs that are mainly expressed in tongue and skin, was related to muscle contraction and keratinocyte differentiation. Functional roles for Module #5 proteins Tead2, Bach1, Notch3 and Dlx5 in the nervous system have reported; this suggests that the other member of Module 5, Fosl2, may have the similar function.

Similarly, Zfp655 in Module #12 was tightly co-expressed with Hnf1a, Hnf1b, Hnf4a and Nr1h4, all of which play important roles in liver (Fig. 4d). This finding suggested that Zfp655, whose function is unknown, may play important roles in liver function. We next identified genes with mRNA expression pattern similar to the Zfp655 DNA-binding activity pattern in the 23 tissues and carried out gene set enrichment analysis (GSEA). The GSEA terms indicated that Zfp655 was positively associated with both

metabolism-related pathways and the immune processes (Fig. 4e), consistent with the functions of Hnf1a/b, Hnf4a and Nr1h4 (ref. 32) (Supplementary Fig. 4a,b).

We also found tightly co-expressed TF pairs, for example, Clock/Arntl (Bmal1), which form a core component of the circadian clock and are widely expressed in several tissues, particularly in circadian tissues[33], with a correlation coefficient was 0.92. Mef2 and Bhlha15 (Mist1) have been reported to play diverse roles in skeletal and cardiac muscle. Mist1 mainly represses MyoD[34], a major TF that regulates muscle differentiation, whereas Mef2a and MyoD can cooperatively activate muscle genes[35]. In our study, Mist1 and Mef2a expression had a negative correlation coefficient of 0.70. Consistently, Mist1 has been reported to be an important Xbp1 target gene (TG) that inhibits muscle differentiation[36], and Xbp1 was negatively correlated with Mef2a ($-0.68$) (Supplementary Fig. 4c). These positive/negative correlations confirmed the collaborative/antagonistic relations of the TFs in biological processes.

**Combinatorial TF network in mouse tissues.** Combinatorial TF interactions are critical for cellular functions and are important determinants of different cell types. The TF interaction network constructed from protein–protein interaction assays, such as the mammalian and yeast two hybrid assay[22,37], have revealed many important regulatory features of TFs. One limitation of such heterologous assays is that the over-expressed proteins may or may not be co-expressed in the same cells. TF atlas allows us to survey the combinatorial TF interactions among different tissues from the perspective of endogenously expressed proteins.

We re-analysed the TF interaction network of the 24 adult tissues by matching differential TF expression patterns to the TF network map[22] (Fig. 5a and Supplementary Data 5). Similar to the tissue specificity of TF expression, TF–TF interactions exhibited an L-shaped distribution among tissues (Fig. 5b). Thus, 764 tissue-restricted TF–TF interactions were identified in no more than 12 tissues, and 252 ubiquitous TF–TF interactions were identified in more than 12 tissues (Supplementary Data 5).

Ubiquitous TFs, such as Jun, Smad3 and Rxra, were involved in large number of TF–TF interactions implying that they are involved in wide variety of cellular transcriptional programs through protein–protein interactions. In contrast, ttrTFs, such as Nr1h4, Irf4 and Ptf1a, have fewer interaction partners, suggesting these TFs may function in tissue-restricted processes (Supplementary Fig. 5a). The positive correlation between number of tissues expressed and the number of TF–TF connections was significant (Fig. 5c).

Interactions between ubiquitous TFs and not ubiquitous TFs were widely observed throughout TF network, which occurred more frequently than expected and were higher than interactions within each groups (ubiTFs–ubiTFs, non-ubiTFs–non-ubiTFs) (Fig. 5d). As exemplified by the connection between Meis1 and Homobox the ubiquitous TF Meis1 was expressed in 23 out of 24 adult tissues and has the potential to connect to 15 ttr-Homobox TFs (Fig. 5e). The specificity of the interaction between Meis1 and Homobox TFs appeared to be determined by the specificity of the Homobox TF. Thus, TF–TF interactions are more conserved among tissues than the TFs themselves. A signature TF networks of each tissue type were also constructed (Supplementary Fig. 5b), and TFs that were expressed in multiple tissues formed different sub-TF networks. For example, Hnf4a interacted with Smad, Hnf1a and Nr2c2 in the stomach, small intestine and colon, whereas it was also connected to Esr1 in the liver, white adipose tissue (WAT) and kidney (Supplementary Fig. 5c).

Our analysis also revealed similar TF expression patterns between adjacent tissues in the nervous system (brain, eyes and spinal cord), digestive system (stomach, colon, intestine and liver), adipose tissue (brown adipose tissue (BAT) and WAT) and immune system (thymus, spleen and blood) (Fig. 5f). We identified top 30 differentially expressed TFs for each of the ten physiological systems (Figs 1a, 5g and Supplementary Data 5) and analysed their downstream TG. Our results revealed that they were significantly enriched in the predominant biological processes of the corresponding systems (Supplementary Fig. 5d).

**TFs that maintain the tissue identity.** We sought to identify TFs that may be required to maintain the identities of the tissue types (tissue-type-maintenance transcription factor (ttmTFs)). We reasoned that ttmTFs should not only be specifically enriched in the tissue but should also dominantly control the transcription of their downstream genes in that tissue. We gathered the mRNA expression data from publically available RNA microarray studies and employed TF-downstream TG data from CellNET[38,39]. A total of 286 ttmTFs were identified in 21 adult tissues, ranging from 62 in thymus and 4 in pancreas, and none in thyroid and seminal vesicle (Fig. 6a,b, Supplementary Fig. 6 and Supplementary Data 6).

Notably, a number of ttmTFs identified here were consistent with previously reported roles of directly converting fibroblasts into the major cell type of the tissue (Table 1). For example, Hnf4a, Hnf1a and Foxa2 had been reported to drive the direct conversion of fibroblasts to hepatocytes[9,40] and Myt1l, Lhx3 and Isl1 have been shown to convert MEF to spinal motor neurons[41]. These activities indicate dominant roles that ttmTFs play in determining tissue identity.

The identification of ttmTFs of tissues provides a conceptual framework for understanding how tissue identity is maintained by TFs. It is logical to predict that genes controlled by two or more ttmTFs may represent the biological processes of particular tissues relatively specifically. We submitted the TGs that were co-regulated by 2 ttmTFs to Reactome Pathway Database[42,43] to find these processes. Reactome terms that were enriched in TGs controlled by dual ttmTF represent the major functions of the tissue (Fig. 6c). For example, TGs of multiple ttmTFs in eye, brain and spinal cord were enriched in neuron-related items, whereas TGs related to immune functions were enriched in spleen and thymus (Supplementary Data 6).

All tissues are developed from three germ layers: ectoderm, mesoderm and endoderm. Because adult tissues are terminally differentiated, they cannot be clearly clustered into their germ layer origins based on their global proteome. We performed principal component analysis (PCA) to test whether the TF patterns were more suitable for clustering tissues to the 'germ' layers. We obtained clustering accuracy of 75% with all TFs and 80% with ttmTFs (Fig. 6d), outperforming the accuracy calculated with TF mRNA expression profiling[22] The correlation between the ttmTFs and 'germ layers' exceeded 0.7 and was larger than those of the non-ttmTFs and all TFs (Fig. 6e).

As ttmTFs play central roles in regulating gene expression in tissues, we investigated the upstream regulatory factors control-ling ttmTFs in tissues. DNA methylation is an epigenetic mark that is critical for mammalian development and tissue lineages, and tissue-specific differentially methylated regions (tsDMRs) occur at distal cis-regulatory elements. These 'vestigial' enhancers are hypomethylated and lack active histone modifications in adult tissues but nevertheless exhibit activity during embryonic development, suggesting that epigenetic memory of embryonic development may be retained in adult tissues[44]. The ttmTF promoters are highly enriched in tsDMR compared with those of

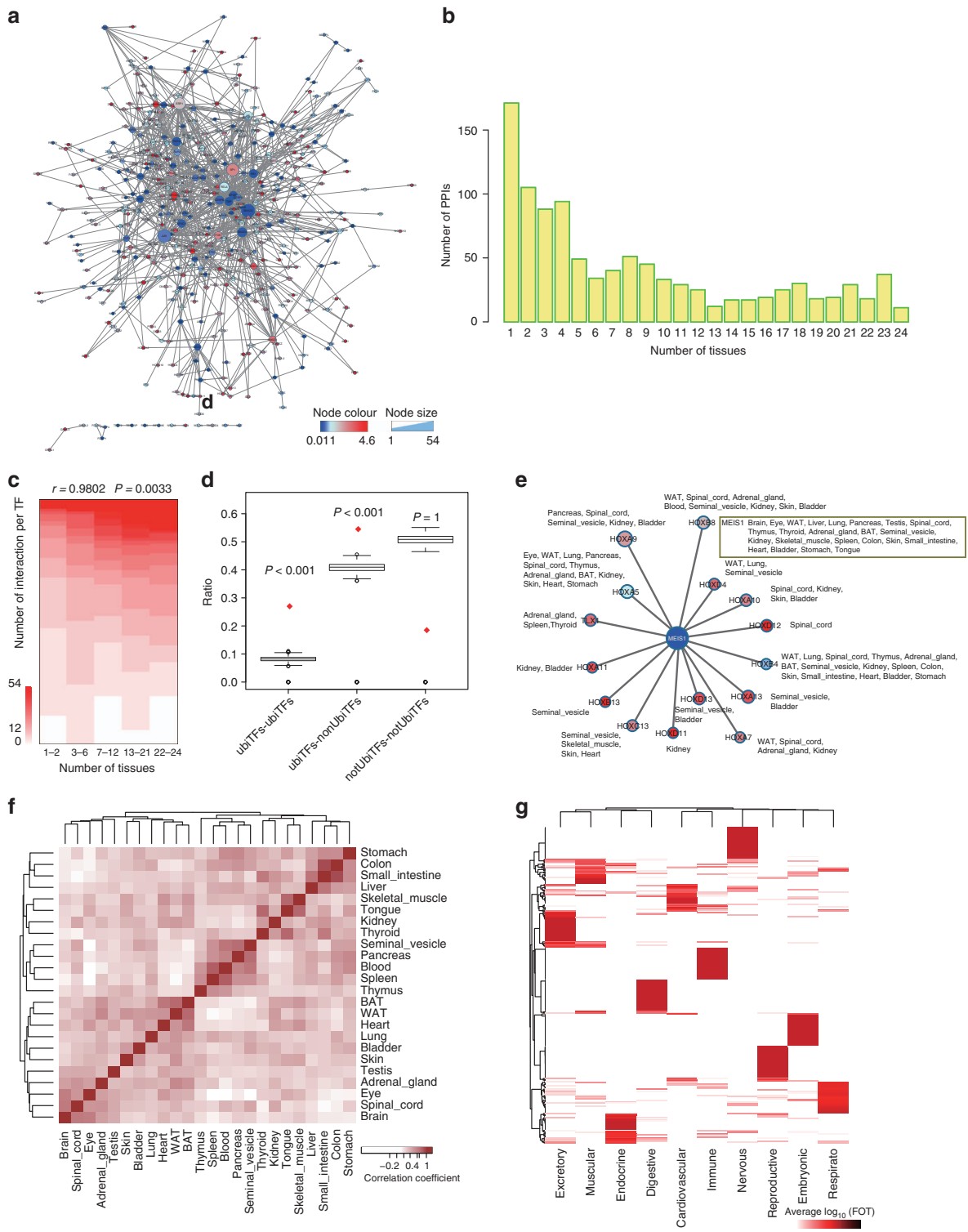

**Figure 5 | TF interaction network and specific TF expression patterns in physiological systems in mouse tissues.** (**a**) Interaction network of all TFs in the 24 tissues. The colours indicate the tissue-specificity score (TSPS), and the node size shows the number of PPIs of the TF. (**b**) The numbers of TF–TF interactions that were detected in different number of tissues. More than 150 TF–TF interactions were detected in one tissue and only 11 TF–TF interactions were detected in all 24 tissues. (**c**) The negative correlation between tissue specificity and the number of TF–TF connections. The TFs were binned into five groups of approximately equal size based on tissue specificity (x axis). The stacks of coloured segments represent the number of interactions for each bin. (**d**) Statistical significance of different types of protein–protein interaction, calculated using 1,000-time permutation test. Centre line and box limits represent median value and lower or upper quartile, respectively. Whiskers show the range of lower quartile −1.5-fold IQR to higher quartile +1.5-fold IQR. (**e**) TF interactions between Meis1 and Homobox TFs in different tissues. As a facilitator hub, Meis1 interacts with different TFs in the Hox family in different tissues, showing the specificity of the interaction was determined by Homobox TFs, namely the ttr TFs. (**f**) Heat map showing the pairwise correlations between all 24 adult tissues based on DNA-binding ability. The TF expression patterns are similar between tissues of the same physiological systems. (**g**) Clustering of the top 30 most enriched TFs in the ten physiological systems to identify TFs that can specify the biological systems.

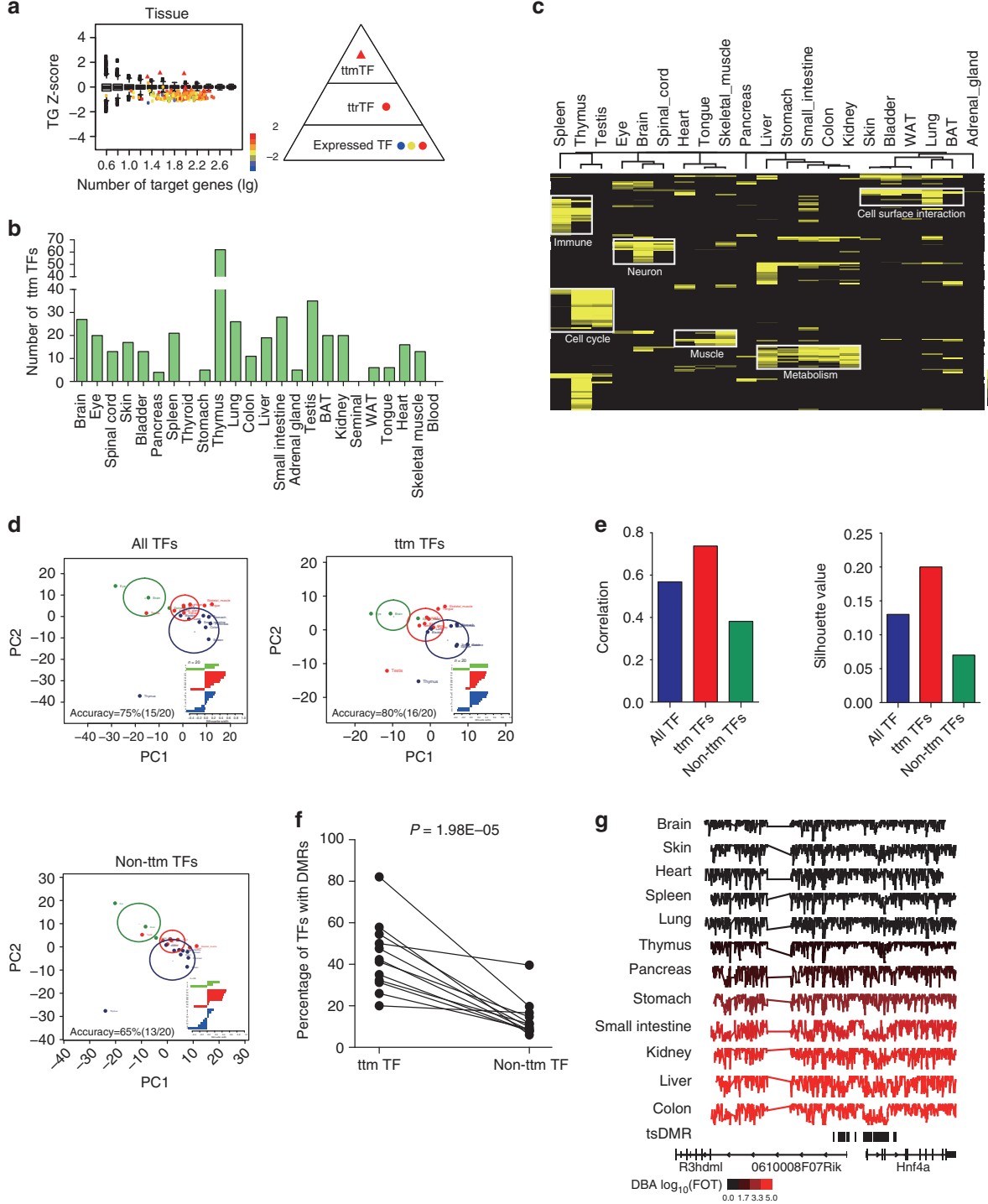

**Figure 6 | Identification of tissue-type maintenance transcription factors (ttmTFs) and their fundamental features.** (**a**) ttmTFs for each tissue are identified with high Z-scores (>1) and TG Z-scores that are higher than those from random data. x axis represents the number of target genes; y axis represents the average TG Z-scores. Network of TFs with a three-tiered organization was shown on the right. (**b**) The number of identified ttmTFs in the 24 adult tissues. (**c**) Heat map for the Reactome terms that were enriched in the TGs that were regulated by at least two ttmTFs in each adult tissue. (**d**) PCA demonstrated that the ttmTFs exhibited the highest accuracy in separating the tissues from the ectoderm, mesoderm and endoderm lineages. Plus signs indicate the cluster centroids. The radius of the circle indicates the average distance from the centroid. The silhouette value was used to measure separation between the three tissue classes. Accuracy was defined as the ratio of tissues with silhouette value more than zero. Green: ectoderm; Red: mesoderm; Blue: endoderm. (**e**) The correlation between PC1 and 'blastoderm' information (left) was the highest using ttmTF set. The silhouette value (right) was used to measure how similar an object is to its own cluster (cohesion) relative to the other clusters (separation). (**f**) The ratios of promoters to the tissues' tsDMRs for both ttmTFs and non-ttmTFs. The 1,000-time permutation test was performed. (**g**) DNA methylation maps near the *Hnf4a* locus in 12 adult mouse tissues. Each track spans percent mCG (%mCG) values between 0 and 100%. The tracks include two tissues from ectoderm, four tissues from mesoderm and six tissues from endoderm. The tissues were ordered according to Hnf4a DNA-binding activities (increasing). The colour bar indicates the TF DNA-binding activities in log$_{10}$(FOT).

**Table 1 | Overlap of ttmTFs with known TFs involved in iPS cell transforming.**

| Species | TFs | Starting cells | Target cells | Reference |
|---|---|---|---|---|
| Human | OCT4 | Human dermal fibroblasts | Blood progenitors | 50 |
| | ASCL1, LMX1A and NR4A2 | Human fibroblasts | Dopaminergic neurons | 51 |
| | OCT4, **SOX2**, **KLF4** and **c-MYC** | Human fibroblasts | Endothelial cells | 52 |
| | **GATA4**, MEF2C and **TBX5** | Cardiac fibroblasts | Cardiomyocytes | 53 |
| | **SOX2** | Fetal foreskin fibroblasts | Neural stem cells | 54 |
| | PRDM16 and **CEBPB** | Skin fibroblasts | BAT | 55 |
| | **NEUROD1**, ASCL1, **BRN2**, MYT1L, LHX3, HB9, **ISL1** and NGN2 | Fetal foreskin fibroblasts | Spinal motor neurons | 41 |
| Mouse | Ascl1, **Brn2** and Myt1l | MEFs, TTFs | Neurons | 56 |
| | **Sox2**, Foxg1 and **Brn2** | MEFs | Neural precursor cells | 57 |
| | **Hnf4a** + Foxa1, **Foxa2** or Foxa3 | MEFs, dermal fibroblasts | Hepatocytes | 58 |
| | Ascl1, Lmx1a and Nurr1 | Mouse fibroblasts | Dopaminergic neurons | 51 |
| | Nr5a1, **Wt1**, **Dmrt1**, Gata4 and **Sox9** | MEFs | Embryonic Sertoli-like cells | 59 |
| | **Gata4**, Mef2c and **Tbx5** | TTFs | Cardiomyocytes | 53 |
| | **Sox2** | MEFs | Neural stem cells | 54 |
| | Ngn3, **Pdx1** and Mafa | Pancreatic exocrine cells | Islet β-cells | 60 |
| | Gata4, **Hnf1a** and **Foxa3** | TTFs | Hepatocytes | 40 |
| | Prdm16 and **Cebpb** | Skin fibroblasts | BAT | 55 |
| | Ngn3 | Hepatic progenitor cells | Neo-Islets | 61 |
| | **Sox10**, **Olig2** and Zfp536 | MEFs | Oligodendroglial cells | 62 |
| | Ascl1, **Brn2**, Myt1l, Lhx3, Hb9, **Isl1** and Ngn2 | MEFs | Spinal motor neurons | 41 |

The ttmTFs are marked in bold. BAT, brown adipose tissue; MEFs, mouse embryonic fibroblasts; TTFs, tail-tip fibroblasts.

non-ttmTFs (Fig. 6f, Supplementary Fig. 7a and Supplementary Data 6). For example, *Hnf4a* locus was hypomethylated in the digestive system, particularly in the liver and colon, compared with other tissues (Fig. 6g).

**Regulation of liver TF pattern in response to perturbations**. We asked whether the TF hierarchy would change and particularly ttmTF regulation, under physiological and pathological conditions. We employed liver regeneration induced by partial hepatectomy (PHx) that removed ~70% of the liver as a model for perturbation. As PHx is a dynamic process and prone to experimental variations, we included more initial experimental conditions to define a more accurate initial state. We combined 26 data sets of liver TFs obtained by catTFRE in physiology conditions and use them as the control experiment to construct a liver TF reference map that defines abundance range for each TF (Supplementary Data 7).

The catTFRE experiments identified 420 TFs for the liver regeneration process, of which 401 TFs could be quantified and compared with the TF reference map of the liver. We found that the intensities of 188 TFs after PHx were greater than the upper quartile values (Q3) of the TF reference map in at least two out of the three individual measurements, and the 188 TFs were defined as outliers. We grouped TFs into four representative stages of liver regeneration according to their temporal behaviour: priming (0–12 h), early progression (24–48 h), later progression (3–5 days) and termination (5–7 days). The TFs related to immune response and NF-kB activation were immediately stimulated within 12 h; TFs that regulate developments constituted the second wave. Cell cycle signalling were activated in the progression phase and downregulated in the terminating phase when the Wnt/beta-catenin pathway was repressed and TGF-beta-Smad pathway was upregulated (Fig. 7a).

One third of the 188 outlier TFs can be classified into the four major functions categories, namely immune and stimulus response, development and differentiation, nuclear receptors and metabolism, repressors and brakes. Immune and developmental TFs are overexpressed in the earlier/middle stage, whereas nuclear receptor and repressors were dominant in the middle/later stage (Fig. 7b).We found that Myc/Max/Mad behaved as a switch in liver regeneration. Myc was stimulated in the priming stage while its antagonist Mad was unregulated in the terminating stage, suggesting the Myc/Max/Mad network is at work in regulating liver regeneration (Supplementary Fig. 7c).

We found that ttmTFs tended to be downregulated when the fate of the tissue was altered. The liver ttmTFs were significantly decreased compared with that of the non-ttmTFs after PHx (Fig. 7c and Supplementary Fig. 7d), indicating liver cells lost their homeostasis and identity or dedifferentiation when dramatic perturbation occurred. Six members of the hepatocyte nuclear factor family in the liver ttmTF group were markedly downregulated in 12 h and 3 days after PHx, and showed a tendency of returning to their original and stable state in the terminating phase (Fig. 7d).

Profiling of the global liver proteome during PHx identified downregulated proteins, which could be used as the TG products to trace back TFs that regulate them (Supplementary Fig. 7e). Many of these TFs that controlled downregulated proteins during PHx were indeed ttmTFs (Supplementary Fig. 7f and Supplementary Data 7). These results suggested that the liver lost ttmTFs during the process of tissue expansion—a trait that normally is not ascribed to liver, and implicated the importance of losing ttmTFs when the identity of the organ is perturbed during liver regeneration.

**Discussion**
Here, we present an atlas of mouse TF DNA-binding activities from 24 adult tissues and 8 fetal tissues (www.tfatlas.org) In the data set, an average of 290 TFs was identified per tissue, and more than 60% of the TFs were encoded in the mouse genome, ensuring that the TF atlas provides a comprehensive global view.

TFs are considered relatively low-abundance proteins in the proteome; however, we found that the TF sub-proteome spanned almost seven orders of magnitude in abundance, revealing a large variation in the abundance of TFs. More than half of the detected TFs were enriched in a certain tissue, and few non-specific or ubiquitous TFs were identified.

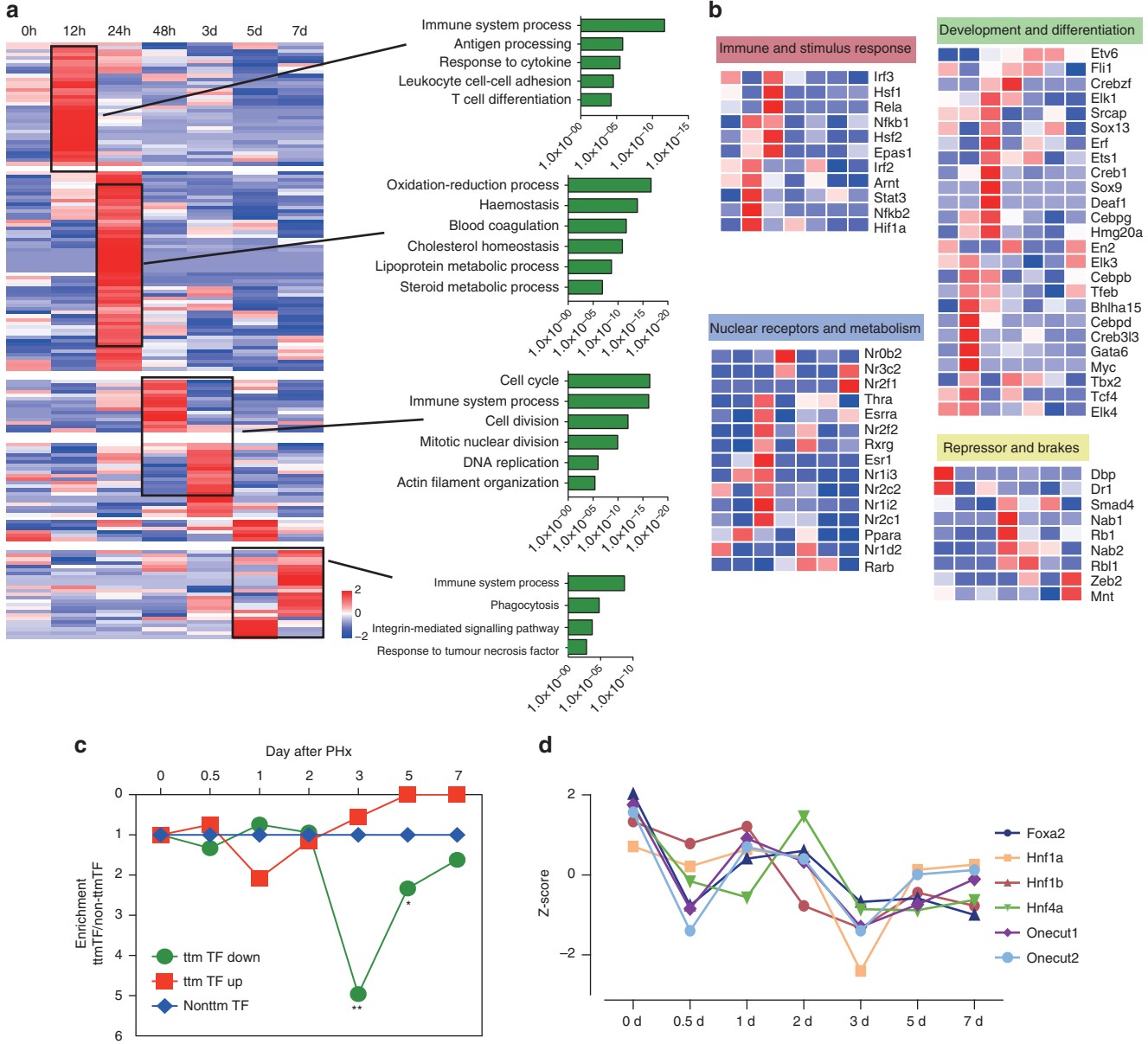

**Figure 7 | Landscape of TF dynamics after PHx. (a)** Four major upregulated TF groups and their accordingly GO pathways in different stages during liver regeneration. GO terms were enriched using the TGs that were regulated by at least two TFs in the sub-clusters and expressed in according PHx data. **(b)** Four major functions categories of outlier TFs, namely immune and stimulus response, development and differentiation, nuclear receptors and metabolism, repressors and brakes. **(c)** Enrichment of downregulated and upregulated ttmTFs at each time point after PHx. The ttmTFs were significantly downregulated. The levels of downregulated and upregulated non-ttm TFs were used as the baseline. The $P$ values were calculated using Fisher's exact test, $*P < 0.1$, $**P < 0.01$. **(d)** Dynamic change of six hepatocyte nuclear factors in ttmTF group during liver regeneration.

TFs often interact with each other to regulate gene transcription[22]; these combinatorial interactions are commonly identified by the yeast two-hybrid method. Using the endogenous TF expression data in different tissues, these combinatorial interactions among TFs derived from the artificial over-expression system in yeast could be verified, such that an interaction between two TFs could be ruled out if they were not co-expressed in the same tissue. Interestingly, TFs with high specificities tend to have less connectivity, whereas ubiquitous TFs engage in more interactions than other TFs. As the largest node, Smad3 was expressed in 23 out of the 24 adult tissues and was connected with 38 TFs, ranging from the Smad3–Max interaction in 23 tissues to the Smad3–Pax8 interaction in the kidney alone. These observations suggested that a ubiquitous TF

interacts with different specific TFs to expand its regulatory repertoire and perform regulatory functions in different tissues.

The TF sub-proteome is the driving force that shapes the tissue proteome. We presented different ways of using our uniformed data set of TF atlas for correlative bioinformatics analysis. ttmTFs, which usually compose a small part of the total TFs in a tissue, are the dominant drivers in forming and maintaining tissue identities. ttmTFs are crucial in maintaining tissue identity, and their TGs, particularly the groups that were controlled by multiple ttmTFs, directly indicate the tissue functions. The ttmTFs listed in this study covered most of the published key TFs that were essential in direct tissue/cell type conversion from fibroblast to the major cell type of the tissue. Interestingly, most ttmTFs are among the most highly abundant TFs in the tissue,

indicating that they play a dominant role in the tissue and may be candidates for tissue engineering and regenerative medicine research.

As an *in vitro* approach and being naked DNA template, the catTFRE approach does have its limitations. As naked DNA was used to measure the potential DNA-binding activities of TFs, the REs in the cell may be blocked in a nucleosome context or by other histone/DNA modifications. The 'activity' of a TF as measured by catTFRE may not reflect its actual activity in all loci on the chromosome. While the multiple TFREs on catTFRE allows for the enrichment and identification of many TFs, this approach cannot distinguish the individual TF transcriptional machinery nor dissect the TC–TF complex, prohibiting the construction of the regulatory network between TFs and TCs. A more precise approach, for example, using single TFRE to purify TF–TC complexes may be applied to further validate functional modules drawn in this study.

Taken together, our study constructed an atlas of TF DNA-binding activities in mouse tissues, expanding the existing knowledge of TFs on the proteome wide scale. The TF expression patterns will have implications for our understanding of how TFs execute their regulatory functions in tissues. Our study under-scores the value of monitoring the TF sub-proteome of tissues to uncover the transcriptional determinants of physiology and pathology in mammals and provides a data set that can be used to illuminate the 'dark proteome' of TFs.

## Methods

**Animals and organ collections.** Six–eight-week-old male C57BL/6 mice were purchased from Beijing HFK Bioscience Co., LTD (Beijing, China). All mice were housed under standard conditions (a specific pathogen-free, temperature-controlled micro-environment with a 12-h day/night cycle). Twenty-four adult mouse organs and tissues (eye, skin, bladder, blood, thyroid, seminal vesicle, WAT, BAT, tongue, pancreas, colon, small intestine, stomach, spinal cord, liver, heart, kidney, brain, lung, muscle, spleen, thymus, adrenal gland and testis (males)) were used in this study. Fetal tissues were collected from pregnant mice at different time points (1.5 days, 6.5 days, 13.5 days and 18.5 days after conception) during pregnancy (Fig. 1a). The mice were killed by cervical dislocation. Whole organs were removed, and samples were quick-frozen in liquid $N_2$ and stored at $-80\,°C$ for RNA extraction. Fresh tissue samples were collected to prepare nuclear extracts (NEs). This study received ethical and scientific approval in compliance with the animal care regulations of Institutional Animal Care and Use Committee, National Center for Protein Sciences (The PHOENIX Center, Beijing).

**Preparation of protein TFRE samples.** The tissues were washed twice with ice-cold phosphate-buffered saline to remove blood and other contaminates, then suspended in 800 μl of Cytoplasmic Extraction Reagent I (CER I) buffer (NE-PER kit, Thermo Scientific) and homogenized using a tissue grinder. Nuclear proteins were extracted in accordance with the manufacturer's instructions. Protein concentrations were determined using the Bradford method (Bio-Rad SmartSpec Plus, Bio-Rad Laboratories, Inc., USA). Approximately 0.15–8 mg of the nuclear proteins was extracted from each adult and fetal tissue. In each biological replicate, tissues from 3 to 4 mice were pooled for each sample to further minimize the individual differences between mice.

**catTFRE pull-down and trypsin digestion.** DNA was synthesized by Genscript (Nanjing, Jiangsu Province, China). Biotinylated catTFRE primers were synthesized by Sigma. Dynabeads (M-280 streptavidin) were purchased from Invitrogen. Approximately 2–3 pmol of biotinylated DNA was pre-immobilized on Dynabeads and then mixed with NEs from the tissues. The mixtures were supplemented with ethylenediaminetetraacetic acid/ethylene glycol-bis(β-aminoethyl ether)-N, N,N′,N′-tetraacetic acid to a final concentration of 1 mM, adjusted with NaCl to a total salt concentration of 200–250 mM, and then incubated for 2 h at 4 °C. The supernatant was discarded, and the Dynabeads were washed twice with NETN (100-mM NaCl, 20-mM Tris-Cl, 0.5-mM ethylenediaminetetraacetic acid and 0.5% [vol/vol] Nonidet P-40) and then twice with phosphate-buffered saline. The catTFRE pull-down beads were re-suspended with 20 μl of 1× SDS loading buffer and boiled for 5 min. The samples were then loaded on 10-cm 10% SDS–polyacrylamide gel electrophoresis gels and run to 1/3 of the length. The gel was stained with Coomassie brilliant blue and then destained in 5% ethanol/10% acetic acid solution. Six bands were excised according to the molecular weight ranges and then subjected to in-gel trypsin digestion, as previously described[45].

**Partial hepatectomy.** All partial hepatectomy surgeries were performed between 8:00 and 10:00. The mice were fasted for 12 h before surgery and intra-abdominally anesthetized with pentobarbital sodium (30 mg kg$^{-1}$). The PHx mouse model (two-thirds hepatectomy) was established as described by Mitchell and Willenbring[46]. After 12, 24, 48, 72, 120 and 168 h, the PHx mice underwent relaparotomy, and liver tissues were taken for the TFRE and profiling experiments. Groups of animals ($n = 3$–$5$ for each group) were killed at 0, 0.5, 1, 2, 3, 7 days after surgery (Supplementary Fig. 7b) and two biological repeats were done with an additional two operational repeats for one biological repeats, making a total of three measurements per time point.

**TF quantification capability.** We performed serial dilution experiment with 3 pmol of catTFRE DNA with different amount of total NE (nuclear exact, 200 μg, 500 μg, 1, 2 and 5 mg) from brain tissue (Supplementary Fig. 1a,b). The total MS signal of TFs (chromatographic peak area) has high correlation coefficient ($R^2 = 0.959$) with the total NE amounts. Notably, we got an excellent linearity when NE amount ranged from 1 to 5 mg. To this end, we used 3 pmol of DNA and 2 mg of total NE for screening the TF atlas of mouse tissues.

**LC-MS/MS analysis using Q exactive plus and orbitrap fusion.** The samples from catTFRE in-gel digestion were analysed on a Q Exactive Plus MS (Thermo Fisher Scientific) interfaced with an Easy-nLC 1,000 nanoflow LC system (Thermo Fisher Scientific). Tryptic peptides were dissolved with 10 μl of loading buffer (5% methanol and 0.2% formic acid), and 5 μl was loaded onto a homemade trap column (2 cm) packed with C18 reverse-phase resin (particle size, 3 μm; pore size, 120 Å; SunChrom, USA) at a maximum pressure of 280 bar with 12 μl of solvent A (0.1% formic acid in water). Peptides were separated on a 75 μm × 15 cm silica microcolumn (3 μm C18, homemade) with a linear gradient of 5–35% Mobile Phase B (acetonitrile and 0.1% formic acid) at a flow rate of 350 nl min$^{-1}$ for 75 min. The MS analysis was performed in a data-dependent manner with full scans (m/z 400–1,500) acquired using an Orbitrap mass analyser at a mass resolution of 70,000 at m/z 400. Up to 20 of the most intense precursor ions from a survey scan were selected for MS/MS and detected by the Orbitrap at a mass resolution of 15,000 at m/z 400. All the tandem mass spectra were acquired using the higher-energy collision dissociation (HCD) method with normalized collision energy of 27%. The automatic gain control for full MS was set to 3e6, and that for MS/MS was set to 5e4, with maximum ion injection times of 60 and 80 ms, respectively. Dynamic exclusion time was 18 s, and the window for isolating the precursors was 3 Th.

The PHx profiling samples were resuspended in 20 μl of loading buffer and analysed by LC-MS/MS on an Orbitrap Fusion MS using the same nanoflow LC system, column and gradient described above. For data-dependent acquisitions using the Orbitrap Fusion, one MS full scan was performed using the Orbitrap (m/z range = 300–1,400; R = 120,000; target value = 5e5), and then, the most intense ions were selected in top-speed mode for fragmentation by HCD (target value: 5,000; isolation window: 1.6 Th; threshold: 5e3) and MS/MS scans in the IonTrap (IT). The MS/MS spectra were acquired with Rapid Ion Trap Scan Rate. The dynamic exclusion time was 18 s, and the normalized collision energy was set to 32%.

**Peptide identification and protein quantification.** Raw files were searched against the mouse refseq protein database (27,414 proteins, version 04/07/2013) with Proteome Discoverer (Thermo Fisher Scientific, version 1.4) using the MASCOT[47] search engine with percolator[48]. The mass tolerance of the precursor ions was set to 20 p.p.m.. For the tolerance of the product ions, QE Plus was set to 50 mmu, and Fusion was set to 0.5 Da. Up to two missed cleavages were allowed for protease digestion, and the minimal required peptide length was set to seven amino acids. Carbamidomethylation of cysteine was set as a fixed modification, and N-terminal protein acetylation and methionine oxidation were set as variable modifications. For the TFRE experiments, the phosphorylation of Ser, Thr and Tyr residues was also set as a variable modification. The data were also searched against a decoy database so that protein identifications were accepted at a false discovery rate of 1%. Homemade software was used to estimate the protein quantities based on the precursor area under the curve. The amount of each gene product was estimated using a label-free, intensity-based absolute quantification (iBAQ) approach[49]. We then used the fraction of total (FOT) to represent the normalized abundance of a particular TF. FOT is defined as a TF's iBAQ divided by the total iBAQ of all identified proteins. We replaced the FOTs less than $10^{-8}$ with $10^{-8}$ to adjust extremely small values. Then FOTs were multiplied by $10^8$ and log10 transformed to make the minimum value of log10(FOT) transformed to zero.

**TF classification.** We combined all DBPs, including TFs and TCs, to calculate the ratio of DNA-related proteins in the total protein identifications. The results showed DNA-related proteins account for an average of 22% in all protein identifications (ranging from 13.6% (Blood) to 38% (Fetal Brain_E18.5)) and about 10% in all protein abundance (ranging from 2% in Blood to 23% in fetal brain_E18.5)). We defined ubiquitous TFs as TFs detected in more than 50% tissues (12 tissues). Because the separation line for ubiquitous TFs and non-ubiquitous TFs from Fig. 2f was 0.5, we chose the median log10 transformed

value of 0.5 as the threshold. Ubiquitous-non-uniform TFs had a median log10 transformed value $> 0.5$ and a maximum value of $> 10 \times$ the median value. Ubiquitous-uniform TFs had a median log10 transformed value $> 0.5$ and a maximum value of $< 10 \times$ the median value. Tissue-specific TFs only expressed in one tissue not in the others. TFs with a transformed median expression value of $< 0.5$ and without much restriction of the maximum value were non-ubiquitous TFs (non-ubiTFs, 640 TFs, accounting for 71.3% of the identified TFs), which exhibited high expression in only a few tissues; TFs with a transformed median expression value of $> 0.5$ were considered ubiquitous TFs (ubiTFs, 257 TFs, 28.7%) expressed in a wide variety of tissues. Among them, the expression of 27 TFs exhibited a maximum value of less than ten times the median value, indicating a ubiquitous-uniform distribution (27 TFs, 3.1%); the rest of the TFs can be classified as ubiquitous-non-uniform (230 TFs, 25.6%), with a maximum expression value exceeding ten times the medium value.

To identify system-specific TFs, we identified the particular tissues that belong to the same system. An average is taken for individual TFs among the tissues in the system. For the ten systems, a Z-score is computed for all TFs in each system and the top 30 TFs with the largest Z-score in each system were selected as system-specific TFs. The ttmTFs are defined if they satisfied the following two conditions (1) specifically enriched in the tissue with Z-scores $> 1$ and (2) average Z-scores of the mRNA expression level of their TGs exceed that of randomly selected genes. Target genes' mRNA expression data were obtained from the GEO database.

**Protein–protein interaction annotation.** We collected the known protein–protein interactions between TFs from the open-access Human Protein Reference Database (HPRD) (release 9, 2010-04-13). Homemade software was used to analyse the protein–protein interactions among the TFs in each tissue, and each TF was annotated with the number of edges and the TSPS[22]. The TSPS uses relative entropy to quantify the extent to which the observed TF expression pattern departs from the null distribution of uniform expression across all tissues. According to this definition, a minimal TSPS of 0 would be reported for TFs that are expressed uniformly across all tissues, whereas a maximal TSPS of 5 would be reported for TFs that are only expressed in a single tissue. Using the TSPS, two distinct TF populations were separated: one population of TFs with widespread tissue expression (TSPS $< 1$) and a second, smaller population with higher tissue specificity (TSPS $> 1$). The TFs with TSPS $> 1$ were almost the same as the non-ubiquitous TFs defined by us above. A 1,000-time permutation was used to test if interactions between ubiquitous TFs and not ubiquitous TFs were more frequent than expected by chance.

**Quantitative reverse-transcription polymerase chain reaction (RT-qPCR).** Total RNA was purified from tissues using the TRIzol reagent (Invitrogen) according to the manufacturer's protocol. Total RNA samples were verified to be free of contaminating genomic DNA. RT–PCR was performed using the SuperScript III First-Strand Kit (Invitrogen). Real-time PCR was conducted using the SYBR Green PCR Master Mix (TOYOBO) in a Bio-Rad CFX96 Sequence Detection System. The primers were chosen based on the principle described by Timothy and Harukazu et al.[22]. Gene-specific primer pairs were collected from previously published sequences or designed using Primer3 software (http://frodo.wi.mit.edu/primer3/). The primers were purchased from Generay Biotech (Shanghai) Co., LTD. (Shanghai, China), and the sequences are listed in Supplementary Table 2. Each measurement was performed in triplicate, and the results were normalized to the expression of the Actin reference gene.

**mRNA profiling and mapping of methylated DNA regions.** We collected the tissue-specific mRNA expression data from public data sets (brain, eye and seminal vesicle from GSE9954; thymus, thyroid and tongue from GSE1133; BAT, WAT, adrenal gland, bladder, heart, small intestine, kidney, liver, lung, pancreas, skeletal muscle, spinal cord, spleen, stomach, testis and uterus from GSE10246; and integrated data for colon and skin from CellNET). We only used genes with probe sets in all three platforms (GPL1261: Affymetrix Mouse Genome 430 2.0 Array; GPL1073: GNF1M, a non-commercial microarray; and CellNET: a mixed expression profile for tissues constructed from hundreds of microarrays under different conditions). DNA methylation maps for 12 adult mouse tissues were downloaded from GSE42836 near the Hnf4a locus. Moreover, tsDMRs were also downloaded from a previous publication[44].

**Hierarchical clustering and PCA.** Unsupervised clustering was performed using the pheatmap (Pretty Heatmaps) function in the R package (pheatmap, version 1.0.8). Briefly, Pearson's correlation values were calculated for each TF pair based on their tissue distribution profiles. The distances between the rows or columns of a data matrix were computed as the Euclidean distance. The 'Nuclear Receptor Ring of Physiology' illustrated in Fig. 4c was drawn using the application Phylodendron. Separations were determined using PCA. For the ttmTF set, the first principal component resulting from the analysis (PC1) was the main direction that was informative for the blastoderm layer, and PC3 was the main direction that was informative for the system. For all TF sets and non-ttmTF sets, PC2 was the main direction that was informative for the blastoderm layer. The silhouette value was used to measure how similar an object was to its own cluster (cohesion) relative to

the other clusters (separation). Using the first three PCs, the average silhouette value for the ttmTFs was the highest among the three feature sets.

**Co-expression modules.** The correlation coefficient, which represents the occurrence of each TF across all tissues as a vector using coordinates of the TF FOT and the coefficient between each pair of FOT vectors, was calculated according to the Pearson correlation coefficient. If the two vectors did not pass Kendall's Tau test, the coefficient was set to 0. We searched for co-expression modules through diagonal. Three TFs along the diagonal were used as module seeds, and they were required to exhibit good correlation with each other and a correlation coefficient larger than 0.4. A new TF was added to the module if it correlated well with the TFs in the module (that is, if more than 2 coefficient were larger than 0.4). We used GSEA to construct a matrix of the association of the DNA-binding ability of each TF with each of 5,296 gene sets (including 186 Kyoto Encyclopedia of Genes and Genomes (KEGG) gene sets, 217 Biocarta gene sets, 674 Reactome gene sets, 3,396 functional gene sets and 823 gene ontology (GO) biological process terms). Module function was annotated by enrichment analysis using the leading RNAs in the correlated GSEA terms.

**Quantitative liver TF reference.** As PHx is a dynamic process and prone to experimental variations, we decided to include more initial experimental conditions to define a more accurate initial state. We used 26 catTFRE experiments that detected more than 150 TFs with at least one unique peptide in mouse liver as reference. The amount of each TF was quantified using iBAQ described above. FOT for TF and TC was used for normalization. Reference range of each TF was calculated as 25th percentile to 75th percentile. Outlier TF in PHx experiments was defined as those with amount greater than the third quartile (Q3) in two out of three repeated experiments.

**Function annotation.** The GO terms that were enriched in the sets of enriched genes were determined using the Database for Annotation, Visualization and Integrated Discovery (DAVID) Bioinformatics Resource v 6.7 with Fisher's exact test. A Reactome-based pathway enrichment analysis was performed using the Reactome Pathway Analysis tool http://www.reactome.org/#PathwayAnalysisDataUploadPage. 'Project to human' option was used, which denotes that all non-human identifiers in the sample were mapped to their human equivalents before the analysis was performed.

**Data availability.** All raw data and the Mascot output tables have been deposited to iProX and can be accessed with the accession IPX00081900.

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

## Acknowledgements

This work was supported by the Ministry of Science and Technology of China (Grant 2016YFA0502500); National Program on Key Basic Research Project (973 Program, 2014CBA02000); National International Cooperation Grant (2014DFB30010, 2012DFB30080 and CPRIT RP110784); National High-tech R&D Program of China (863 Program, 2014AA020201 and 2015AA020108); National Natural Science Foundation of China (Grant 31270822); National Institute of Health (Illuminating Druggable Genome, Grant U01MH105026) and a grant from the State Key Laboratory of Proteomics (Grant SKLP-YA201401).

## Author contributions

Conceptualization, C.D., M.L. and Q.Z.; Methodology, M.L., T.G. and Q.Z.; Formal analysis, X.X., C.D., Q.Z., J.F. and W.L.; Investigation, J.Q., C.D. and M.L.; Data curation, Q.Z. and M.L.; Visualization, X.X., Q.Z. and Y.L.; Writing—Original draft, C.D., Q.Z. and X.X.; Writing—Review and editing, J.Q., C.D., Y.W. and Q.Z.; Funding acquisition, C.D., B.Z. and J.Q.; Resources, M.L. and Q.Z.; Supervision, C.D. and J.Q.

## Additional information

**Competing interests:** The authors declare no competing financial interests.

