## [Peer Review File · Nature Communications]

Reviewers' comments:

Reviewer #1 (Remarks to the Author):

In this manuscript, Zhou use a DNA-affinity approach to capture and identify proteins that bind to DNA designed to contain a large number of transcription factor (TF) response elements. They do so in 32 mouse tissues, aiming to generate TF profiles for each individual tissue, but more importantly to correlate these to each other to create a hierarchy of TFs that are unique to some or shared between several organs. To this end, the authors invoke a range of bioinformatic analyses to group and classify their data. Specifically, they link TF patterns to tissue functionality, also making use of existing data of TF-TF interactions and TF target genes.

This is a nice study, using a methodology previously developed by the same authors to identify TFs that bind to a DNA construct containing a large number of concatenated TF-response elements. They now applied this to a large number of tissues to rank and classify TF patterns and correlate this with tissue function. The manuscript is well-structured, clearly written, using sound analysis methods, and the figures are of high quality. The limitation of the paper resides in the fact that is mainly a descriptive study based on a single (although large) data set, where only in the last section the authors try to get closer to functionality by monitoring changes in TF profiles after liver regeneration. Yet, this does not lead to a concrete set of TFs that could be conclusively/causatively linked to this process.

Other remarks:

1. One of the limitations of the used approach is that it is an in vitro method, with a number of associated shortcomings. The authors should address these, and put this in the perspective of biologically interpreting their data. For instance, the procedure starts from a nuclear extract, thus taking TFs out of their physiological context (e.g. being in its chromatin-bound or soluble state, which is a dynamic equilibrium for many TFs). Second, the bait is a piece of naked DNA, i.e. devoid of nucleosomes or other chromatin constituents. Therefore, capture of a particular TF (or failure to do so) does not have a direct biological meaning. As a result, the set of TFs that is identified in the end provides a fingerprint that may be used for certain classification procedures, however it does not indicate expression level in the respective tissues, or propensity to be associated with chromatin in vivo. The authors should emphasize this to make the reader aware of this.

2. The authors refer to their method as revealing TF 'DNA-binding activities' and 'abundance/expression level' (e.g p. 10 last paragraph). Although they use these terms almost interchangeably, neither of them is strictly correct, since there are many reasons why the identified proteins are not a representative sampling of the in vivo situation. First, DNA-binding only applies to interaction with naked DNA in an in vitro experiment, as mentioned above. Second, no data are provided with regard to the fraction of each TF that is captured (which may be different for each TF), and to what extent protein binding to the bait-DNA reaches saturation. These factors combined make it very difficult to make any quantitative statement on either DNA-binding activity or protein abundance in tissue.

3. To circumvent some of these issues at least in part, the authors should have overlaid their

data with a proteomic study (PMID 23436904) profiling global protein expression across 28 mouse tissues. Given the very high similarity between the source material in both studies this should provide complementary data where expression levels of at least the most abundant TFs can be properly addressed.

4. It is unclear which TF response elements were included, and how many of the identified proteins corresponded with these.

5. In addition to the above, the authors only mention the number of TFs that were identified, and not the total number of proteins that were co-isolated (as interaction partners or contaminants).

6. The clustering of nuclear receptors (Fig 4a) results in a slightly different classification than previously proposed (page 9). However these groupings are not mutually exclusive as they are assigned by function and location, respectively.

7. Zfp655 was shown to be associated with metabolism and immune-related processes (p11, fig 5e) claimed to be similar to Hnf1a/b, Hnf4a and Nr1h4. However, the latter was not demonstrated and should be included to allow a direct comparison to what is shown in fig 5e.

8. The authors mention that 'expression of the liver ttmTFs was significantly decreased compared with that of the non-ttmTFs' (page 16). They should mention some examples explicitly, and discuss how they are related to liver regeneration. Currently, this is a very general statement that does not provide a lot of insight.

9. The liver is a highly heterogeneous tissue. Can the effects observed in the liver regeneration experiment be ascribed to any particular cell type?

10. I have some hesitation about the terminology described as 'TF hierarchical networks among the tissues' (p. 16). Hierarchy implies interdependence, which does not apply to the large majority of tissues. Instead, repeated identification of the same proteins may indicate re-use of TFs for different functionalities, which, from an organism point of view, may be a necessity given the large variety of constituent cell types (hundreds) and the limited number of TFs encoded in the genome ('just' hundreds).

Reviewer #2 (Remarks to the Author):

In this paper the authors employ the catTFRE approach recently developed by their group to identify the repertoire of TFs active in the nucleus of 24 adult and 8 fetal mouse tissues. This approach, that is based on TF enrichment using tandem repeats of binding sites and mass spectrometry, is a great improvement from whole cell/tissue proteomics as a large fraction of TFs are below the detection limit in these methods, and from expression profiling as there is low correlation between TF mRNA levels and TF activity. This dataset, for which the authors generated an easy to navigate web page, will certainly be useful for the scientific community and serve as a framework to study tissue-specific gene regulatory networks. Having said this, there are several analyses that need to be modified or added to support the several conclusions made by the authors to be acceptable for publication. Limitations of the method should also be discussed.

Major concerns:

- 1) Some of the numbers referenced in the manuscript don't match with those in the webpage. For instance, in lines 139-140 the authors mention 173 and 447 TF for skeletal muscle and thymus, respectively. However, in the TF Atlas webpage these numbers are 167 and 448. Where are the discrepancies coming from?
- 2) The authors should comment on any potential bias in TF abundance due to the method (for instance the tandem motif sequences used in the pull down, affinity, saturation, etc). Two TFs from the same family could bind to the same sequence so their relative FOTs could be influenced by their relative protein abundance and affinity for the motif. Further, TFs from different families may compete with a different number of TFs for sites, so their relative FOTs may also be influenced by number of family members that recognize a motif. From their previous paper, it is clear that using different DNA sequences in the pull down results in different enrichments and abundances. The authors should find a way of controlling for this factors or clearly specify these limitations.
- 3) Is the number of TFs detected in different tissues a property of the number of TFs active in the different tissues or could be explained by differences in the amount of protein in the nuclear extracts?
- 4) Some of the thresholds used seem arbitrary. For instance, why look at the 35 most abundant proteins (line 151), 12 tissues (line 166), a median expression >0.5 (line 172), or 10 times the median (line 178)? A rationale for the thresholds should be included.
- 5) In line 155 the authors "assume" that TFs that are more abundant "regulate the busy and important functions of the tissues." However, there are other factors that influence that such as affinity, which genes they regulate (it could be few but important), etc. The authors could hypothesize this and then show based on data.
- 6) The statement in line 193-194 is not necessarily true. There could be widespread transcriptional regulation by ubiquitous TFs or by a few highly expressed/active TFs.
- 7) Paragraph 200-206 seems disconnected with the rest and there is no conclusion. In addition, in line 202 the authors say that TSG have more ubiquitous tissue distribution, however in figure 3e that difference is not significant. The authors should either make a clear point or remove the paragraph.
- 8) In lines 293-295 the authors state that clusters can shed light on the "dark proteome". However, they do very little effort to validate their predictions besides the example of zfp655. The authors should go beyond the anecdotal example. For how many TFs you can make

functional predictions? For how many of those is the function known in the literature? Does it match?

9) In line 332 the authors reference papers studying protein-protein interactions between TFs using Y2H. However, in reference 13 mammalian two-hybrid is used instead (correct also in line 500). Other references also use other techniques. Literature should be properly cited.

10) In paragraph 356-370 the authors comment on the connection between ubiquitous TFs and ttrTFs, and then list a number of examples. To make this claim the authors need to do statistical analyses. Are interactions between these 2 classes of TFs more frequent than expected by chance?

11) In lines 381-384, how were the TGs determined? This paragraph is unclear.

12) The criteria used to define ttmTFs is not very stringent as many TFs not involved in maintaining tissue identity can be enriched in a tissue and also be coexpressed with its targets. Indeed, the authors classify 30% of the TFs they detect as ttmTFs which seem high. Besides providing some anecdotal examples, the authors should attempt a more systematic analysis to support their claim.

13) Lines 421-425 are impossible to understand. It is also speculative as there is no experiment or analysis showing or suggesting causality between ttmTF concentration and function.

14) Some sentences in the Discussion section are purely speculative, and no evidence is provided in the paper. For instance, lines 492-494, 495-498, 516-518. Overstatements should be avoided.

15) The authors should comment on the limitations of the method in the Discussion section.

Minor concerns:

1) Line 49: "TFs interacting with the promoters of..." Enhancers and silencers also play an important role in gene regulation.

2) Paragraphs lines 62-87: Other methods that study TFs and GRNs should also be mentioned such as yeast one-hybrid assays (PMID 25910213, 23917988), genome-wide DNase footprints (PMID: 22955618), etc.

3) The authors filter the proteins they detect by mass spectrometry based on DBDs. To have a sense of the specificity of the approach, the authors should also mention, at least in the methods section, which proportion of the proteins they detect (in number and in abundance) correspond to TFs.

4) Line 130: DBTF is not defined.

5) In line 142: FOT is not defined.

6) In line 155 the authors mention nuclear receptors (NRs) but in line 133 they talk about NHRs. Consistency should be kept throughout the manuscript.

7) What is the difference between ttrTFs and ubiquitous-non-uniform TFs? Some of the definitions are confusing and there are many acronyms in the paper making it hard to read.

8) Some figures lack appropriate labels, and larger fonts would benefit reading. Figure 2f: MaxValue and MedianValue of what? Figure 3c needs a label in the y-axis. Figure 4a, 4d, 6c, 6e, 6f, 7c need a label for the color gradients. What are the axis in figure 5a? Figure 5e: label missing in top graph. Figure 6f: what is it being clustered? A label is missing in the y-axis of fig 7g.

9) In line 223 the authors say they detected 47 NRs from 32 tissues. But in the following

sentence they talk about half of adult tissues (24 in total). This is confusing.

10) In paragraph 277-284 the authors use cosine similarity. The way it is defined is not very intuitive and it doesn't scale linearly with the overlap in the set of tissues shared by two TFs. The authors should explore other more intuitive measures of similarity such as the Jaccard index or PCC.

11) In paragraph 312-325 the authors mention the correlation coefficient for the expression of TF pairs. Is this based on catTFRE or mRNA expression? Why do the authors use PCC in this case and cosine for figure 5a?

12) Line 352-354: The correlation coefficient and the p-value should be included.

13) Line 382: TG is not defined.

Reviewer #1:

In this manuscript, Zhou use a DNA-affinity approach to capture and identify proteins that bind to DNA designed to contain a large number of transcription factor (TF) response elements. They do so in 32 mouse tissues, aiming to generate TF profiles for each individual tissue, but more importantly to correlate these to each other to create a hierarchy of TFs that are unique to some or shared between several organs. To this end, the authors invoke a range of bioinformatic analyses to group and classify their data. Specifically, they link TF patterns to tissue functionality, also making use of existing data of TF-TF interactions and TF target genes.

This is a nice study, using a methodology previously developed by the same authors to identify TFs that bind to a DNA construct containing a large number of concatenated TF-response elements. They now applied this to a large number of tissues to rank and classify TF patterns and correlate this with tissue function. The manuscript is well-structured, clearly written, using sound analysis methods, and the figures are of high quality. The limitation of the paper resides in the fact that is mainly a descriptive study based on a single (although large) data set, where only in the last section the authors try to get closer to functionality by monitoring changes in TF profiles after liver regeneration. Yet, this does not lead to a concrete set of TFs that could be conclusively/causatively linked to this process.

Q1: One of the limitations of the used approach is that it is an in vitro method, with a number of associated shortcomings. The authors should address these, and put this in the perspective of biologically interpreting their data. For instance, the procedure starts from a nuclear extract, thus taking TFs out of their physiological context (e.g. being in its chromatin-bound or soluble state, which is a dynamic equilibrium for many TFs). Second, the bait is a piece of naked DNA, i.e. devoid of nucleosomes or other chromatin constituents. Therefore, capture of a particular TF (or failure to do so) does not have a direct biological meaning. As a result, the set of TFs that is identified in the end provides a fingerprint that may be used for certain classification procedures; however it does not indicate expression level in the respective tissues, or propensity to be associated with chromatin in vivo. The authors should emphasize this to make the reader aware of this.

Reply: Indeed, catTFRE is an in vitro method and has the limitations mentioned by the reviewer mentioned. In order to demonstrate the feasibility and accuracy of catTFRE approach in dissecting the endogenous TF activity and biological features in proteome scale, we have performed the following procedures and evaluations:

1. While we acknowledge that catTFRE is an in vitro binding method, our previous data demonstrated that our approach is able to monitor the biological response of TF dynamic changes. For instance, we utilized the catTFRE approach to analyze dynamic changes of global TF-DNA binding patterns after TNF- α treatment studies (*Proc Natl Acad Sci U S A.* 2013, PMID: 23553833). Consistent with previous studies, TFs in NF- κ B family and JNK/P38 pathways were activated within 15 minutes (**Figure CL1**). Also, catTFRE pull-downs in K562 cells treated with phorbol myristate acetate (PMA) or imatinib mesylate (Gleevec), demonstrated that the catTFRE can correctly identify the dynamic of the TF activity patterns (**Figure CL1**). More recently (*Anal. Chem.*, 2016, DOI: 10.1021/acs.analchem.6b03150), we performed the catTFRE approach to measure the dynamics of TF patterns in response to EGF treatments. The activation of 14 representative TFs were detected by catTFRE including many well-characterized responders of EGF, such as FOS, JUN, NR4A and MYC. The above data indicated that the catTFRE is able to dissect cellular signaling pathways in the form of transcription factors DNA binding activity changes.

Figure CL1 Systematical and quantitative analysis of TF profiling after TNF- α and EGF stimulation.

A: Kinetic TF activation pattern of 293T cells after TNF- α stimulation. (*Proc Natl Acad Sci U S A.* 2013, PMID: 23553833)

B: Bioinformatics analysis of TF regulations induced by drugs. Functional classification of altered TFs in PMA (left) and Gleevec (right). Down-regulation groups are indicated in blue and up-regulation groups are in brown. (*Proc Natl Acad Sci U S A.* 2013, PMID: 23553833)

C: Dynamic of TF patterns in HeLa cells treated with EGF and temporal profiles of the representative 14 TFs induced by EGF. (*Anal. Chem.*, 2016, DOI: 10.1021/acs.analchem.6b03150)

2. We agree that naked DNA does not represent the natural state of DNA in a living cell as compared to a nucleosome template. We have investigated the difference between naked DNA and nucleosomes in our previous paper in *Molecular Cell* (PMID: 23850489). We

performed DNA-pulldown with naked DNA or nucleosome assembled with core histone octamers and tested them on CoR-ER α -ERE complex formation. On nucleosomal EREs, we were able to detect 16 of the 18 CoRs seen on the naked EREs (Figure CL2). The main effect of the nucleosomal DNA seemed to decrease the amount of TFs bound on the DNA, and thus decrease the signal in mass spectrometry. This makes sense as nucleosomes are known to inhibit TF DNA binding. Considering that we want to construct a TF atlas in mouse tissues with the deep TF coverage, we used naked DNA instead of nucleosomes. In addition to our studies, utilizing naked DNA as bait to survey protein-DNA interactions was a conventional method, which was widely used in many publications (*Proc Natl Acad Sci U S A.* 2013, PMID: 23388641, *Cell Rep.* 2013 PMID: 24139795 and *Cell.* 2011, PMID: 22153072).

To make the readers aware of these limitations, we added the statement above and the references in the discussions.

Figure CL2 CoRs from MCF-7 and HeLa S3 NEs bound with E2-liganded ER α on naked 4xERE-E4 and 4xERE-E4 assembled with Hela core octamers. (*Mol Cell.* 2013, PMID: 23850489)

A: MS identifies at least 17 CoRs from MCF-7 and HeLa S3 NEs. Number of identified peptides,

peptide amount, and fold E2-change (as a heatmap with color scale defined below; same scale is used in later figures) are indicated.

B: MS data (in heatmap format for fold change in the presence of HeLa core histones) show that HeLa core nucleosomes (+hist) decrease binding of CoR-ERE complexes from MCF-7 NE.

Q2: The authors refer to their method as revealing TF ‘DNA-binding activities’ and ‘abundance/expression level’ (e.g p. 10 last paragraph). Although they use these terms almost interchangeably, neither of them is strictly correct, since there are many reasons why the identified proteins are not a representative sampling of the in vivo situation. First, DNA-binding only applies to interaction with naked DNA in an in vitro experiment, as mentioned above. Second, no data are provided with regard to the fraction of each TF that is captured (which may be different for each TF), and to what extent protein binding to the bait-DNA reaches saturation. These factors combined make it very difficult to make any quantitative statement on either DNA-binding activity or protein abundance in tissue.

Reply: To eliminate the ambiguity, we uniformly use “TF DNA-binding activity” in the revision.

To evaluate the quantification capability of catTFRE, we measured the saturation curve of catTFRE. We performed serial dilution experiments with 3 pmol of catTFRE DNA (the exact amount used throughout the study) with different amount of NE (200ug, 500ug, 1mg, 2mg and 5mg) from mouse brain tissue. As shown in the **figure CL3**, the total MS signal of TFs (chromatographic peak area) has high correlation coefficient ($R^2=0.959$) with the total NE amounts. Notably, an excellent linear response was obtained when NE amount ranged from 1mg to 5mg. Based on these results, we used 3 pmol of DNA and 2mg of total NE for screening the TF atlas of mouse tissues.

We also surveyed individual TFs in the dilution experiments and found good linear response characters (**Figure CL4**).

Figure CL3 Quantitative feasibility and linearity of catTFRE strategy evaluated by dilution analysis. Different amounts of NE extracted in brain were used as shown. Total peptide AUC (area under curve) was calculated.

Figure CL4 Quantitative feasibility and linearity of individual DBTFs. Total peptide AUC from 20 TFs selected in Figure CL3 were calculated.

Taken together, we have demonstrated that the catTFRE approach can sensitively and accurately monitor the abundance and DNA-binding activity dynamics of TFs with dilution and many “proof

of principle” experiments. Also, we evaluated the saturation curve to set up optimized conditions for DNA pull-down MS pipeline. Please see **Supplementary Fig. 1** in the revision.

We appreciate the reviewer’s comment for precisely pointing out the shortcomings of the catTFRE. We added this limitation to the discussion section in the revision.

Q3: To circumvent some of these issues at least in part, the authors should have overlaid their data with a proteomic study (PMID 23436904) profiling global protein expression across 28 mouse tissues. Given the very high similarity between the source material in both studies this should provide complementary data where expression levels of at least the most abundant TFs can be properly addressed.

Reply: We compared our catTFRE dataset with Geiger et al.’s protein profiling dataset (*Mol Cell Proteomics*. 2013, PMID: 23436904). It is clear that catTFRE is able to capture more TFs compared with protein profiling (941 TFs in our dataset vs 151 TFs by Geiger et al.). The 10 TFs that were exclusively identified in the Geiger et al.’s dataset were observed in special development stages of tissues, which were not included in our study.

Figure CL5 Venn diagram shows our data covers most of TFs that Geiger et al. identified.

As expected, TFs detected by proteome profiling tend to be the high abundant ones in the catTFRE dataset (**Figure CL6**). We also calculated correlation coefficients for the 13 overlap tissues in both datasets (**Table CL 1**) and found that the Spearman’s rank correlation coefficient

ranged from 0.046 (liver) to 0.401 (spleen), suggesting a poor correlation between TF expression levels and their DNA-binding activities.

Tissue	Matched median rank	Not matched median rank	p value
Adrenal gland	236	165	0.000122389
BAT	186	139.5	0.002056854
Colon	196	136.5	8.13E-05
Eye	241	196	0.052411871
Heart	133	137	0.172295335
Liver	229	154	0.000645924
Lung	204	156	0.002623138
Skeletal muscle	120	86	0.377651493
Pancreas	220	136.5	1.09E-05
Spleen	258	160.5	2.17E-08
Stomach	218	148.5	0.000401787
Thymus	314.5	208.5	9.83E-07
WAT	205	144.5	0.001188382

Figure CL6 TFs detected in profiling data are in higher abundance part of TFRE data. Y-axis showed TF rank in TFRE data. Red boxes are overlapped TFs by profiling data and TFRE data. P value was calculate using Wilcoxon rank-sum test.

Table CL1 Comparison of dataset with catTFRE approach to Geiger et al.'s dataset with profiling approach. Correlation coefficient was calculated by Spearman's rank correlation analysis.

Tissue	#TF (catTFRE)	#TF (Geiger et al)	Overlap	Correlation coefficient
Adrenal gland	357	85	72	0.179
Brown fat	297	66	53	0.227
Colon	305	94	74	0.272
Eye	403	70	51	0.314

Heart	272	62	43	0.131
Liver	316	80	59	0.0463
Lung	326	77	60	0.277
Muscle	173	32	14	0.147
Pancreas	295	62	49	0.327
Spleen	346	76	61	0.401
Stomach	311	51	45	0.273
Thymus	407	86	76	0.246
White fat	307	77	54	0.116
Total	804	138	134	

Furthermore, we mined hundreds of published literatures (**Supplementary Data 1** in the revision) to examine some of the “important” TFs in the 13 overlapped tissues. As showed in **table CL2**, the catTFRE approach detected most of the “important” TFs in tissues (76 out of 85), while the profiling data only identified few of them (6 out of 85).

Table CL2 Summary of tissue “important TFs” in the 13 overlap tissues between the two datasets.

Tissue	Important TF	catTFRE	Geiger et al	Overlap
Eye	10	9	0	0
Lung	5	5	0	0
Pancreases	9	5	0	0
Adrenal gland	4	3	0	0
Thymus	8	6	1	1
Heart	10	10	0	0
WAT	7	3	1	1
Liver	8	8	3	3
BAT	6	4	0	0
Colon	4	4	1	1
Muscle	4	2	0	0
Spleen	6	6	0	0
Stomach	4	4	0	0

In summary, comparison between the catTFRE and the profiling datasets indicated that the catTFRE could more accurately monitor the TF binding activities and represent the biological features of endogenous TFs in the tissues. We acknowledge that the comparison is not entirely fair as the mass spectrometry technique has made great advancement and the profiling data was collected with last generation mass spectrometer. Nevertheless, we added the comparison between catTFRE and profiling results in the revision and pointed out the differences in technology used. Please see **Supplementary Fig. 1** and **Supplementary Data 1** in the revision.

Q4: It is unclear which TF response elements were included, and how many of the identified proteins corresponded with these.

Reply: We referred to TF binding database JASPAR to select consensus TFREs for different TF families. To design the catTFRE construct, we used 100 selected TFREs and placed two tandem copies of each sequence with a spacer of three nucleotides in between, resulting in a total DNA length of 2.8 kb. In the mouse TF atlas, we identified 87 identified TFs whose response elements correspond to the designed TFREs. Moreover, we also identified large number of additional TFs whose response elements were not included in the catTFRE sequence. In the previous work (*Proc Natl Acad Sci U S A.* 2013, PMID: 23553833), we have speculated the possible reasons why the number of TFs identified by catTFRE greatly exceeded the original design of 100 TF families: (i) the 3-bp linkers may create additional binding sites; (ii) the tandem TFRE may also create additional binding sites, and (iii) the flexibility of TFs in TFRE recognition.

Q5: In addition to the above, the authors only mention the number of TFs that were identified, and not the total number of proteins that were co-isolated (as interaction partners or contaminants).

Reply: Indeed the catTFRE pulled down many transcriptional co-regulators (TCs) and other DNA binding proteins (DBP). In the mouse TF atlas, we identified 523 TCs, ranging from 63 in skeletal muscle to 366 in thymus (**Table CL3**). **Figure CL7** summarizes the distribution of TCs detected in

the 32 mouse tissues. Similar to TF's pattern, an L-shaped distribution pattern was observed among the 32 tissues. Interestingly, TC showed a lower tissue-specificity score (TSPS) than the TFs ($P = 3.86E-16$), indicating their ubiquitous distribution (**Figure CL8**). Among them, six subunits of Mi-2/NuRD complex (Rbbp7, Hdac1, Hdac2, Mbd2, Rbbp4 and Mbd3) were identified. Mi-2/NuRD is an important protein complex coupling chromatin remodeling ATPase and chromatin deacetylation functions, and plays an essential role in gene expression through epigenetic regulation. We added the description of TCs in **Supplementary Fig. 2** in the revision.

Figure CL7 A: Number of tissues in which the TCs are expressed. B: Comparison of TSPS between TF and TC.

Figure CL8 Heatmap for TCs in the 32 mouse tissues.

Table CL3 The number of DNA binding protein, transcription cofactor, DNA binding transcription

factor and their proportions in 32 mouse tissues.

Tissue	DNA binding Protein	TC	TF	Total*	Total Protein	Ratio (Protein ID)	Ratio (Abundance) [#]
MEF	479	180	224	601	2418	24.9%	11.9%
fBrain_18.5	591	171	366	719	1893	38.0%	23.0%
fBrain_13.5	661	263	371	849	2705	31.4%	17.0%
fLiver_13.5	531	182	261	643	1734	37.1%	17.9%
fLiver_18.5	388	97	185	440	1492	29.5%	12.9%
Uterus_1.5	364	105	177	423	1457	29.0%	13.7%
Embryo_6.5	450	140	232	544	1792	30.4%	13.9%
Placenta_18.5	453	134	229	540	2204	24.5%	11.4%
Brain	593	235	310	754	3669	20.6%	8.1%
Eye	730	261	403	903	3309	27.3%	11.7%
WAT	608	217	307	750	3735	20.1%	6.2%
Liver	601	231	316	762	3827	19.9%	8.3%
Lung	674	260	326	850	4435	19.2%	6.6%
Pancreas	676	282	295	885	5463	16.2%	3.8%
Testis	731	319	339	962	4560	21.1%	7.7%
Spinal cord	721	300	359	947	5235	18.1%	5.6%
Thymus	866	366	447	1138	4865	23.4%	7.3%
Thyroid	415	99	201	474	2629	18.0%	8.1%
Adrenal gland	721	290	357	923	4862	19.0%	6.1%
BAT	603	249	297	780	4344	18.0%	6.2%
Blood	455	199	175	596	4389	13.6%	2.0%
Seminal vesicle	557	245	228	730	4542	16.1%	3.5%
Kidney	731	286	377	928	4629	20.0%	7.7%
Skeletal muscle	346	63	173	380	1704	22.3%	8.2%
Spleen	761	303	346	980	4814	20.4%	6.0%
Colon	697	320	305	918	5620	16.3%	4.2%
Skin	754	279	390	960	4927	19.5%	6.5%
Small intestine	692	263	340	868	3779	23.0%	10.6%
Heart	539	174	272	650	3179	20.4%	8.8%
Bladder	808	339	396	1056	6030	17.5%	6.2%
Stomach	650	264	311	832	5099	16.3%	4.7%
Tongue	426	107	212	488	2825	17.3%	5.1%
Liver_Profiling	126	37	17	152	2175	7.0%	0.1%

* Total includes DNA binding protein, TC and TF; there were some overlap among them.

The ratio of abundance was the amount of TFs in total proteins.

Q6: The clustering of nuclear receptors (Fig 4a) results in a slightly different classification than

previously proposed (page 9). However these groupings are not mutually exclusive as they are assigned by function and location, respectively.

Reply: The slightly different classification between our dataset and previously proposed (*Cell*, 2006, PMID: 16923397) has revealed the diversity and complementary information provided by TF DNA binding activity and gene expression at mRNA level. Also, the similar conclusions between Bookout AL et al.'s work and the catTFRE results further demonstrated that the catTFRE approach could quantitatively detect the TF DNA binding activities at proteome scale and is applicable to study the biological features of the TFs. We have added this discussion in the revision.

Q7: Zfp655 was shown to be associated with metabolism and immune-related processes (p11, fig 5e) claimed to be similar to Hnf1a/b, Hnf4a and Nr1h4. However, the latter was not demonstrated and should be included to allow a direct comparison to what is shown in fig 5e.

Reply: We apologize for not explaining it clearly. The module #12 includes 5 members, Hnf1b, Hnf1a, Hnf4a, Nr1h4, and Zfp655. Using GSEA approach, we examined the correlation coefficient between Zfp655 of TFRE data and gene expression of microarray data and predicted that the function of the Zfp655 were significantly related to metabolism (such as retinol metabolism, PPAR signaling pathway, fatty acid metabolism) and immune process. In the revision, as suggested by the reviewer, we performed the GSEA approach to investigate the function enrichment for the other members in module #12, including Hnf1a, Hnf1b, Hnf4a, and Nr1h4. Similarly, as shown in the **figure CL9**, these 4 members were also significantly associated with retinol metabolism, lipid metabolism, and PPAR signaling pathway, consistent with the previous literatures (*Mol Cell Biol*. 2001, PMID: 11158324 , *Science*. 2004, PMID: 14988562 and *Hepatology*. 2008, PMID: 18972444), suggesting that the TFs in the same co-expression module tend to have similar biological functions. We have added the function enrichment of other 4 members to **Supplementary Fig. 4** in the revision.

Figure CL9. GSEA terms of other four TFs in Module #12 suggest their functions in metabolism and immunity.

Q8: The authors mention that ‘expression of the liver ttmTFs was significantly decreased compared with that of the non-ttmTFs’ (page 16). They should mention some examples explicitly, and discuss how they are related to liver regeneration. Currently, this is a very general statement that does not provide a lot of insight.

Reply: We thank reviewer for the suggestion. In the previous version, we only performed very superficial analysis on the changes of TF patterns in liver regeneration. In the revision, we investigated the TF dynamics in liver regeneration in more details.

As PHx is a dynamic process and prone to experimental variations, we decided to include more initial experimental conditions to define a more accurate reference state. We combined 26 datasets of liver TFs obtained by catTFRE in physiology conditions in our database. These 26 experiments can be considered as the control experiment, and calculated the range for each TF to construct a TF reference map of the liver organ. To find changed TFs during PHX, we identified a total of 188 outlier proteins that were greater than the upper quartile (Q3) of the TF reference map in at least two out of the three biological repeat experiments. We summarized 4 distinct temporal TF functional groups in the 4 representative stages in the process of liver regeneration from day 0.5 to day 7 (12-24h, priming stage; 24-48h, early progression stage; 3-5d, later progression stage; 5-7d, terminating stage). The TF related to immune response and NF-kB

activation pathways were immediately stimulated after partial hepatectomy within 12 hours. TFs regulating development constituted the 2nd wave. Cell cycle signaling were activated in the progression phase and down regulated in the terminating phase when the Wnt/beta-catenin pathway was repressed and TGF-beta-Smad pathway was upregulated (**Figure CL10**). We were able to assign 1/3 of the 188 outliers into four major functions categories, including immune and stimulus response, development and differentiation, nuclear receptors and metabolism, repressors and brakes. Immune and developmental TFs are overexpressed in the earlier/middle, while nuclear receptor and repressors were dominant in the middle/later stage (**Figure CL10**).

Myc/Max/Mad network, which is formed by the oncogene Myc, MYC associated factor X (MAX), and mitotic arrest deficient protein (MAD) regulates gene activation and repression by switching between antagonistic interaction pairs of Myc–Max and Max–Mad. We found that Myc/Max/Mad behaved as a switch in liver regeneration. Myc was stimulated in the priming stage while its antagonist Mad was upregulated in the terminating stage, suggesting that Myc/Max/Mad network plays a role in regulating liver regeneration (**Figure CL10**).

Figure CL10 Landscape of TF dynamics after PHX.

A: Four major up regulated TF groups and their accordingly GO pathways in different stages during liver regeneration.

B: Four major functions categories of outlier TFs, namely immune and stimulus response, development and differentiation, nuclear receptors and metabolism, repressors and brakes.

C: Myc/Max/Mad network in regulating liver regeneration.

Focusing on ttmTFs, we found that expression of the liver ttmTFs was significantly decreased compared with that of the non-ttmTFs after PHx, indicating liver cells lost their identity undergoing drastic perturbations like PHx. Six members of hepatocyte nuclear factor family in liver ttmTF group were markedly down regulated in 12h and 3 days after PHx, and displayed a tendency to return to the original and stable state in the terminating phase (**Figure CL11**).

Figure CL11 Dynamic change of six hepatocyte nuclear factors in ttmTF group during liver regeneration.

In another published study (*Mol Cell Proteomics*. 2016, PMID: 27562671), we monitored the proteome alteration in the whole process of the *ex vivo* culturing of primary hepatocytes. We found that dominant liver functions such as lipid metabolism and drug metabolism were down regulated during the process of *ex vivo* culturing, and the down regulated proteins were mostly the gene products of the ttmTFs of the liver (**Figure CL12**), suggesting that the tissue loses its ttmTFs when the fate is altered. The changes of the ttmTFs in PHx and *ex vivo* culturing revealed a similar feature and regulation of the ttmTFs.

Figure CL12. Downregulated proteins in cultured HC supernatants were enriched in target gene groups of liver ttmTFs. (*Mol Cell Proteomics*. 2016, PMID: 27562671)

We have added the new analysis and the reference above in the revision. Please see **Fig. 7** and **Supplementary Fig. 7** in the revision for more details.

Q9: The liver is a highly heterogeneous tissue. Can the effects observed in the liver regeneration experiment be ascribed to any particular cell type?

Reply: This is a very important issue – cell type resolved proteomics. Indeed, the liver consists of 4 major cell types, including hepatocytes (HCs), hepatic stellate cells (HSCs), Kupffer cells (KCs), and liver sinusoidal endothelial cells (LSECs). In liver regeneration, how the 4 cell types behave is not completely clear and will be the subject of study in the future.

We recently published “A Cell-type-resolved Liver Proteome” on *Molecular & Cellular Proteomics* (PMID: 27562671). In that study, we have purified 4 major cell types from mouse liver and performed deep proteome coverage for each cell type. In-depth proteomics identified 6000 to 8000 gene products (GPs) for each cell type and a total of 10,075 GPs for four cell types. This dataset could serve as an ideal background library to investigate the enrichment of the up-regulated TFs of different stages in the 4 particular cell types. We tentatively assigned cell type enriched TFs from the proteome profiling data of the 4 major liver cell types when expression of a TF in a certain cell type is 3 times above the average value in the 4 cell types. We then mapped the up-regulated outlier TFs with the cell type enriched TFs during PHx. Only 1 to 6 cell-type enriched TFs were observed to be up-regulated in the process (**Table CL4**). It seemed that more LSEC enriched TFs were up-regulated in the process. Since LSEC also expressed more TFs than the other cell types, we normalized the number of up-regulated cell type enriched TFs with the number of TFs detected in the cell and displayed the result in **Figure CL13**. LSEC and KC seemed to be the cell types that displayed more dynamic regulation in the process.

Table CL4 Number of up regulated cell type enriched TFs of different stages, all TFs and cell type enriched TFs in four major cell types.

Number of TFs	HC	HSC	KC	LSEC
All TF	125	279	239	307
Cell type enriched TF	14	39	31	79
0.5d	2	2	4	5
1d	1	1	0	6
2d-3d	2	1	2	5
5d-7d	1	1	2	0

Figure CL13 Normalized ratio of up regulated cell type enriched TFs of different stages in four major liver cell types during liver regeneration. The numbers are the original number of up regulated cell type enriched TFs.

Q10: I have some hesitation about the terminology described as 'TF hierarchical networks among the tissues' (p. 16). Hierarchy implies interdependence, which does not apply to the large majority of tissues. Instead, repeated identification of the same proteins may indicate re-use of TFs for different functionalities, which, from an organism point of view, may be a necessity given the large variety of constituent cell types (hundreds) and the limited number of TFs encoded in the genome ('just' hundreds).

Reply: We agree with the reviewer's point and understand his/her hesitation about using the

word “hierarchical networks”. We used the term for lacking of a better or more precise word. We thus removed the adjective “hierarchical” in the revision.

References:

1. Ding C, Chan D W, Liu W, et al. Proteome-wide profiling of activated transcription factors with a concatenated tandem array of transcription factor response elements [J]. *Proceedings of the National Academy of Sciences*, 2013, 110(17): 6771-6776.
2. Shi W, Li K, Song L, et al. Transcription Factor Response Elements on Tip: A Sensitive Approach for Large-Scale Endogenous Transcription Factor Quantitative Identification [J]. *Analytical Chemistry*, 2016, 88(24): 11990-11994.
3. Foulds C E, Feng Q, Ding C, et al. Proteomic analysis of coregulators bound to ER α on DNA and nucleosomes reveals coregulator dynamics [J]. *Molecular cell*, 2013, 51(2): 185-199.
4. Mirzaei H, Knijnenburg T A, Kim B, et al. Systematic measurement of transcription factor-DNA interactions by targeted mass spectrometry identifies candidate gene regulatory proteins [J]. *Proceedings of the National Academy of Sciences*, 2013, 110(9): 3645-3650.
5. Viturawong T, Meissner F, Butter F, et al. A DNA-centric protein interaction map of ultraconserved elements reveals contribution of transcription factor binding hubs to conservation [J]. *Cell reports*, 2013, 5(2): 531-545.
6. Slattery M, Riley T, Liu P, et al. Cofactor binding evokes latent differences in DNA binding specificity between Hox proteins [J]. *Cell*, 2011, 147(6): 1270-1282.
7. Geiger T, Velic A, Macek B, et al. Initial quantitative proteomic map of 28 mouse tissues using the SILAC mouse [J]. *Molecular & Cellular Proteomics*, 2013, 12(6): 1709-1722.
8. Hayhurst G P, Lee Y H, Lambert G, et al. Hepatocyte nuclear factor 4 α (nuclear receptor 2A1) is essential for maintenance of hepatic gene expression and lipid homeostasis [J]. *Molecular and cellular biology*, 2001, 21(4): 1393-1403.
9. Odom D T, Zizlsperger N, Gordon D B, et al. Control of pancreas and liver gene expression by HNF transcription factors [J]. *Science*, 2004, 303(5662): 1378-1381.
10. Wang Y D, Chen W D, Wang M, et al. Farnesoid X receptor antagonizes nuclear factor κ B in hepatic inflammatory response [J]. *Hepatology*, 2008, 48(5): 1632-1643.
11. Ding C, Li Y, Guo F, et al. A cell-type-resolved liver proteome [J]. *Molecular & Cellular Proteomics*, 2016, 15(10): 3190-3202.

Reviewer #2:

In this paper the authors employ the catTFRE approach recently developed by their group to identify the repertoire of TFs active in the nucleus of 24 adult and 8 fetal mouse tissues. This approach, that is based on TF enrichment using tandem repeats of binding sites and mass spectrometry, is a great improvement from whole cell/tissue proteomics as a large fraction of TFs are below the detection limit in these methods, and from expression profiling as there is low correlation between TF mRNA levels and TF activity. This dataset, for which the authors generated an easy to navigate web page, will certainly be useful for the scientific community and serve as a framework to study tissue-specific gene regulatory networks. Having said this, there are several analyses that need to be modified or added to support the several conclusions made by the authors to be acceptable for publication. Limitations of the method should also be discussed.

Reply: Many thanks for reviewer's positive comments. We have added the limitations of catTFRE approach in the discussion section.

Major concerns:

Q1: Some of the numbers referenced in the manuscript don't match with those in the webpage. For instance, in lines 139-140 the authors mention 173 and 447 TF for skeletal muscle and thymus, respectively. However, in the TF Atlas webpage these numbers are 167 and 448. Where are the discrepancies coming from?

Reply: We apology for making the mistakes when updating the dataset to the website. We have reloaded the dataset and checked manually to ensure the correctness.

Q2: The authors should comment on any potential bias in TF abundance due to the method (for instance the tandem motif sequences used in the pull down, affinity, saturation, etc). Two TFs from the same family could bind to the same sequence so their relative FOTs could be influenced by their relative protein abundance and affinity for the motif. Further, TFs from different families may compete with a different number of TFs for sites, so their relative FOTs

may also be influenced by number of family members that recognize a motif. From their previous paper, it is clear that using different DNA sequences in the pull down results in different enrichments and abundances. The authors should find a way of controlling for this factors or clearly specify these limitations.

Reply: Indeed, the catTFRE may have some shortcomings as the reviewer mentioned. For example, (1) the bindings between TFs and DNA follow the affinity curves, only the optimized experimental condition with the proper TF-DNA molar ratio can better represent the relative TF abundances; (2) the flexibility of TFs in TFRE recognition could result in two TFs from the same family or different families compete with each other in binding the catTFRE.

To avoid the (1) potential bias and competition of TF binding from the same or different families, and (2) the saturation of the TF - catTFRE binding, we performed serial dilution experiment with 3pmol of catTFRE DNA (the exact amount used in the whole study) with different amount of total NE (200ug, 500ug, 1mg, 2mg and 5mg) from brain tissue. As shown in the **figure CL3**, the total MS signal of TFs (chromatographic peak area) has high correlation coefficient ($R^2=0.959$) with the total NE amounts. Notably, we got an excellent linear response when NE amount ranged from 1mg to 5mg. To this end, we used 3 pmol of DNA and 2mg of total NE for screening the TF atlas of mouse tissues. We also surveyed individual TFs in the dilution experiments and found good linear response characters (**Figure CL4**).

Figure CL3 Quantitative feasibility and linearity of catTFRE strategy evaluated by dilution analysis. Different amounts of NE extracted in thymus were used as shown. Total peptide AUC (area under curve) was calculated.

Figure CL4 Quantitative feasibility and linearity of individual DBTFs. Total peptide AUC from 20 TFs selected in Figure CL3 were calculated.

In our previous studies (*Proc Natl Acad Sci U S A.* 2013, PMID: 23553833), a “proof of principal” experiment was applied to validate the sensitivity and accuracy of catTFRE in monitoring the

biological response of TF dynamic changes. Deserve to be mentioned, TFs from the same family behaved quite consistently in response to TNF- α treatment (ATF2, ATF3, ATF7; JUND, JUNB, JUN; NF- κ B, REL, RELA; etc.) under the experimental conditions used.

Figure CL1 Systematical and quantitative analysis of TF profiling after TNF- α and EGF stimulation.

A: Kinetic TF activation pattern of 293T cells after TNF- α stimulation. (*Proc Natl Acad Sci U S A.* 2013, PMID: 23553833)

B: Dynamic of TF patterns in HeLa cells treated with EGF and temporal profiles of the representative 14 TFs induced by EGF. (*Anal. Chem.*, 2016, DOI: 10.1021/acs.analchem.6b03150)

In another work we published recently (*Anal. Chem.*, 2016, DOI: 10.1021/acs.analchem.6b03150), we performed the catTFRE approach to measure the dynamic of TF patterns in response to EGF treatments. The activation of 14 representative TFs were detected by catTFRE including many well-characterized responders. The members from the bzip family such as FOSB, FOS, FOSL1, JUNB, and JUN were up regulated consistently. These results have revealed that with the excessive DNA bait and optimized experimental conditions, the potential biases in TF identification could be significantly eliminated. The catTFRE could accurately monitor the TF

binding activities and represent the biological features of endogenous TFs in the tissues.

Taken together, we have demonstrated that the catTFRE approach can sensitively and accurately monitor the abundance and DNA-binding activity dynamics of TFs with dilution and many “proof of principle” experiments. Also, we evaluated the saturation curve to set up optimized conditions for DNA pull-down MS pipeline.

To make the readers aware of these potential biases, we have emphasized the limitations in the discussion section.

Q3: Is the number of TFs detected in different tissues a property of the number of TFs active in the different tissues or could be explained by differences in the amount of protein in the nuclear extracts?

Reply: In the TF atlas experiments, we uniformly utilized 2mg of total nuclear extracts for catTFRE pulldown. To further explore whether the number of TFs detected was based on the amount of protein (protein number) in different tissue nuclear extracts, we chose brain, liver, and thymus organs and performed the NE profiling and catTFRE experiments. As shown in the **table CL5**, different number of TFs was identified with the same amount of proteins (nuclear extracts) in 3 tissues. And a positive correlation exists between the amount of protein number in NE and the corresponding TF number identified by catTFRE.

Table CL5 Summary of the NE profiling and catTFRE experiments in brain, liver and thymus.

NE source	Protein# in NE profiling	TF# in NE profiling	TF# in catTFRE*	TF# in Atlas
Brain	3097	56	283	310
Liver	2716	50	210	316
Thymus	3247	144	315	447

* TF#s identified from one catTFRE MS run.

Q4: Some of the thresholds used seem arbitrary. For instance, why look at the 35 most

abundant proteins (line 151), 12 tissues (line 166), a median expression >0.5 (line 172), or 10 times the median (line 178)? A rationale for the thresholds should be included.

Reply:

1. As shown in the TF abundance cumulative curve (**Figure CL15**), the top 28 TFs (among the top 3% of TFs in abundance) account for 90% of total TF abundance/mass. In the revision, we chose 28 as the threshold of the high abundant TFs instead of 35. Among top 28 high abundant TFs, Nfia/b/c/x are ubiquitous transcription regulators, Arntl is one of the core circadian TFs in mammals (*Cell. 2014*, PMID: 25416951), Esrra, Nr2f2, Nr2f6 and Rxra are nuclear receptors (NRs) that regulate metabolism and other functions (*Cell. 2006*, PMID: 16923397), Max is an important component in the Myc/Max/Mad TF network that controls cell cycle progression and differentiation (*Curr Opin Genet Dev. 1994*, PMID: 8193530).

Figure CL15 Cumulative curve of TFs' DNA binding activities.

2. As for the number of 12 tissues selected: We used TSPS (tissue-specificity score) to examine TF specificity (*Cell. 2010*, PMID: 20211142). Previous study used the threshold of TSPS ≥ 1 to separate TFs that are wide-spread expressed and TFs with higher specificity. The correlation coefficient between TSPS and number of tissues was -0.93481 ($P < 2.2E-16$). TSPS=1 was about the number of tissues 12, which is half the number of all 24 adult tissues. In the revision, we have explained the details for the selection.

3. As for the number of median 0.5 selected as the threshold for ubiquitous TFs: We defined ubiquitous TFs as TFs detected in more than 50% tissues (12 tissues). For TFs expressed in less than 12 tissues, the median value was 0. For TFs expressed in 12 tissues, median value ranged from 0.16 to 1.02, with most under 0.5. Also the separation line from Fig. 2F was 0.5. So we chose the median value of 0.5 as the threshold for ubiquitous TFs.

4. We chose 10 times the median as threshold for ttrTFs, because it was a stringent threshold for our TF data. Uhlen et al. used 5 fold of the average as the threshold for tissue enhanced genes (*Science*. 2015, PMID: 25613900). Kim et al. used 10 fold as the threshold for detecting fetal-tissue-restricted genes (*Nature*. 2014, PMID: 24870542).

We added the explanation for the setting of thresholds in the revised manuscript.

Q5: In line 155 the authors “assume” that TFs that are more abundant “regulate the busy and important functions of the tissues.” However, there are other factors that influence that such as affinity, which genes they regulate (it could be few but important), etc. The authors could hypothesize this and then show based on data.

Reply: In the revision, we deleted the overstatements or not-so-accurate statements to ensure the integrity of the study. We have deleted this statement “more abundant TFs regulate the busy and important functions of the tissues” in the revision, as it is just an assumption.

Q6: The statement in line 193-194 is not necessarily true. There could be widespread transcriptional regulation by ubiquitous TFs or by a few highly expressed/active TFs.

Reply: Thanks for the reviewer’s suggestion. We have removed the “overstatement” in this version.

Q7: Paragraph 200-206 seems disconnected with the rest and there is no conclusion. In addition, in line 202 the authors say that TSG have more ubiquitous tissue distribution,

however in figure 3e that difference is not significant. The authors should either make a clear point or remove the paragraph.

Reply: We thank the reviewer for the comment. We have removed this paragraph in the version.

Q8: In lines 293-295 the authors state that clusters can shed light on the “dark proteome”. However, they do very little effort to validate their predictions besides the example of zfp655. The authors should go beyond the anecdotal example. For how many TFs you can make functional predictions? For how many of those is the function known in the literature? Does it match?

Reply: As suggested by the reviewer, we expanded the analysis to 22 out of 37 co-expression modules with both number of TFs involved in each module less than 13 TFs and higher tissue specificity (z score>1.5). As indicated in the **Supplementary Data 4**, we predicted potential functions of 22 modules consisting 156 TFs, and 131 of them matched previous reports. The rest of the 25 TFs are not reported before shed the light on the disclosing of the “dark proteome” and deserve further functional evaluation (**Table CL6**). For example, Module #1 containing 6 TFs that are mainly expressed in tongue and skin is related to muscle contraction and keratinocyte differentiation; the functional correlation to nervous system of Tead2, Bach1, Notch3 and Dlx5 in Module #5 was also reported, whereas another member Fosl2 was not reported yet. Module #31 is enriched in brain, eye and spinal cord; its members Nkx2-2, Sox2 and Arnt2 are essential for brain development and the central nervous system.

We have added the details in the revision in **Page 11** and **Supplementary Data 4**.

Table CL6 Predicted functions of 25 TFs in 15 modules.

TF	TF numbers in module (<13)	Tissue Enrich (Zscore>1.5)	Predicted Functions	Module NO.
Tshz2	5	eye thyroid	eye development; establishment of skin barrier	Module #4
Fosl2	5	Brain	nervous system	Module #5

Zfp592	5	thymus; colon; intestine	immune system process	Module #10
Tsc22d4	5	Blood Thymus	immune; haematopoietic	Module #11
Zfp655	5	Liver Kindey colon intestine	metabolism	Module #12
Hoxc5	7	small intestine; colon	cholesterol homeostasis; intestinal absorption	Module #13
Zscan18	6	liver pancreas stomach	metabolism	Module #14
Zscan22	5	liver BAT	lipid homeostasis	Module #15
Zfp219; Zfp451; Hivep1	12	Testis; thymus	cell cycle; immune system process	Module #16
Klf12; Zbtb2; Bhlhe41	7	Eye Thymus	eye development; nervous system	Module #18
Foxl2;Tef; Alx4	12	eye	nervous system	Module #30
Zfp608; Tgif2	6	brain ;testis	hypothalamo-pituitary-gonadal	Module #32
Esrrb	7	eye	neural cells; eye development	Module #36
Zfp533; lkzf2; Zfp668; Zfp672	6	thymus; spleen	immune system process; cell cycle	Module #20
Elk3; Elk4; Six1	10	NA	immune system process	Module #24

Q9: In line 332 the authors reference papers studying protein-protein interactions between TFs using Y2H. However, in reference 13 mammalian two-hybrid is used instead (correct also in line 500). Other references also use other techniques. Literature should be properly cited.

Reply: We apologize for the errors. We have corrected the references and cited the proper literature in the revision.

Q10: In paragraph 356-370 the authors comment on the connection between ubiquitous TFs and ttrTFs, and then list a number of examples. To make this claim the authors need to do statistical analyses. Are interactions between these 2 classes of TFs more frequent than expected by chance?

Reply: We investigated the connection between different TFs sub-groups. As shown in **figure CL16**, the connection between ubiquitous TFs and ttrTFs was more frequent than expected by chance (ratio = 17.5%, $P < 0.001$). We think it is better to calculate statistical significance of

connection between ubiquitous TFs and non-ubiquitous TFs, instead. As shown in the **figure CL16**, connection between ubiquitous TFs and non-ubiquitous TFs was more frequent than expected by chance (ratio = 54.5%, $P < 0.001$) and higher than other two groups (ubiTFs-ubiTFs, non-ubiTFs – non-ubiTFs). We have added this in the revision (**Fig. 5d**).

Figure CL16 Statistical significance of different types of protein-protein interaction. (A) Statistical significance of ubiquitous TFs and ttrTFs interaction. (B) Statistical significance of ubiquitous TFs and non-ubiquitous TFs interaction.

Also, we have updated the item explanation of the TF sub-groups in the Method section.

ttrTFs: tissue type restricted TFs, TFs that are expressed in a particular tissue at levels that are at least 10 times higher than the median value of all adult tissues as tissue type restricted TFs
Tissue specific TFs: TFs only expressed in one tissue and not in any others.

ttrmTFs: a TF that satisfied two characteristics: (1) specifically enriched in the tissue and (2) average Z-scores of TGs' mRNA expression levels exceeding those of randomly selected TGs.

Not Ubiquitous TFs: TFs with a transformed median expression value of < 0.5 and without much restriction of the maximum value were non-ubiquitous TFs.

Ubiquitous TFs: TFs with a transformed median expression value of > 0.5 were considered ubiquitous TFs.

Ubiquitous-not-uniform TFs: Among ubiquitous TFs, the expression of twenty-seven TFs exhibited a maximum value of less than 10 times the median value, indicating a ubiquitous-uniform distribution.

Ubiquitous-uniform TFs: the rest of the TFs can be classified as ubiquitous-non-uniform (230 TFs,

25.6%), with a maximum expression value exceeding 10 times the medium value.

Q11: In lines 381-384, how were the TGs determined? This paragraph is unclear.

Reply: We apologize for not explaining it clearly. The target genes (TG) of TFs were downloaded from the previous literature of CellNet (*Cell*. 2014, PMID: 25126793). The TF-TG regulations were inferred through thousands of mouse gene expression profiling data. We have added the details in the revision in the Materials and Methods section.

Q12: The criteria used to define ttmTFs is not very stringent as many TFs not involved in maintaining tissue identity can be enriched in a tissue and also be coexpressed with its targets. Indeed, the authors classify 30% of the TFs they detect as ttmTFs which seem high. Besides providing some anecdotal examples, the authors should attempt a more systematic analysis to support their claim.

Reply: We nominated ttmTFs of a tissue that should satisfy with two characteristics, (1) specifically enriched in this tissue compared to other tissues; (2) the significantly over-expression of its target gene group in this tissue. With these criteria, we are able to determine TF groups that may have critical roles in a certain tissue. Importantly, one of the hot topic in TF studies in recent years was using particular TFs to convert MEF cells to a specified tissue or major tissue cell types, such as liver, heart, and neuron cells (*Nature*. 2011, PMID: 21716291, *Cell*. 2010, PMID: 20691899 and *Nature*. 2010, PMID: 20107439). These TFs that can be used for tissue direct conversion is called master TFs of a tissue. Our nominated ttmTF list has covered almost half of them known to date (**Table 1**), further revealing the value of the nominated ttmTF list.

We also use other analysis strategy to evaluate the correlation between the ttmTF function and the tissue features. For example, we submitted target genes co-regulated by two ttmTFs for Reactome analysis. Reactome terms that are enriched in dual-ttmTF target genes represent the major function of the tissue (**Fig. 7c**), suggesting ttmTFs that we nominated may carry out essential functions in the tissue.

We apologize for not explaining the number of ttmTFs properly. The ttmTF groups behave quite diverse among different tissue. Even though a total of 286 ttmTF were nominated, accounting for 30% of total identified TFs, the percentage of ttmTF in a particular tissue is low – an average of 18 (only take up 10% of identified TFs) in each tissue.

Q13: Lines 421-425 are impossible to understand. It is also speculative as there is no experiment or analysis showing or suggesting causality between ttmTF concentration and function.

Reply: We have deleted this statement in the revision.

Q14: Some sentences in the Discussion section are purely speculative, and no evidence is provided in the paper. For instance, lines 492-494, 495-498, 516-518. Overstatements should be avoided.

Reply: We removed these overstatements in the revision.

Q15: The authors should comment on the limitations of the method in the Discussion section.

Reply: We have added the limitations of the catTFRE approach mentioned above in the discussion section to make the readers aware of them.

Page 20 in the revision: “We wanted to point out the limitations of the catTFRE approach. The catTFRE is an in vitro approach and the naked DNA template does not fully represent the natural state of DNA in a living cell or tissue. Even though we have demonstrated the sensitivity and accuracy of catTFRE in monitoring biological responses of TFs by the proof-of-principal type of experiments and we have shown that the naked DNA used in catTFRE has advantage in TF identification than the nucleosome, there are caveats. The flexibility of TFs in TFRE recognition could result in two TFs from the same family or different families compete with each other in

binding the catTFRE. Thus, the catTFRE experimental condition should be optimized to use proper NE and DNA incubating ratio to ensure the linearity of the TF abundance quantification. As naked DNA was used to measure the potential DNA binding activities of TFs, and the response elements in the cell may be blocked in a nucleosome context or other histone/DNA modifications, the “activity” of a TF as measured by catTFRE may not work in all loci in the chromosome.”

Minor concerns:

Q1: Line 49: “TFs interacting with the promoters of...” Enhancers and silencers also play an important role in gene regulation.

Reply: Thanks for the comment. We have updated the sentence to “such as DNA-binding TFs interacting with the cis-elements, including promoters, enhancers and silencers, of the genes they activate or repress.”

Q2: Paragraphs lines 62-87: Other methods that study TFs and GRNs should also be mentioned such as yeast one-hybrid assays (PMID 25910213, 23917988), genome-wide DNase footprints (PMID: 22955618), etc.

Reply: Thanks for the suggestion. We have cited these references in the revision.

Q3: The authors filter the proteins they detect by mass spectrometry based on DBDs. To have a sense of the specificity of the approach, the authors should also mention, at least in the methods section, which proportion of the proteins they detect (in number and in abundance) correspond to TFs.

Reply: The catTFRE approach identified quite a lot of transcriptional co-regulators and other DNA binding proteins. We combined all DNA binding proteins (DBP), including TFs and transcriptional co-regulators (TC), to calculate the ratio of DNA related proteins in the total protein identifications. The results showed that DNA related proteins account for an average of 22% in all

protein identifications (ranging from 13.6% (Blood) to 38% (Fetal Brain_E18.5) and about 10% in all protein abundance (ranging from 2% in Blood to 23% in fetal brain_E18.5) with catTFRE approach (**Table CL4**).

Table CL4 The number of DNA binding protein, transcription cofactor, DNA binding transcription factor and their proportions in 32 mouse tissues.

Tissue	DNA binding	TC	TF	Total*	Total Protein	Ratio (Protein ID)	Ratio (Abundance) [#]
MEF	479	180	224	601	2418	24.9%	11.9%
fBrain_18.5	591	171	366	719	1893	38.0%	23.0%
fBrain_13.5	661	263	371	849	2705	31.4%	17.0%
fLiver_13.5	531	182	261	643	1734	37.1%	17.9%
fLiver_18.5	388	97	185	440	1492	29.5%	12.9%
Uterus_1.5	364	105	177	423	1457	29.0%	13.7%
Embryo_6.5	450	140	232	544	1792	30.4%	13.9%
Placenta_18.5	453	134	229	540	2204	24.5%	11.4%
Brain	593	235	310	754	3669	20.6%	8.1%
Eye	730	261	403	903	3309	27.3%	11.7%
WAT	608	217	307	750	3735	20.1%	6.2%
Liver	601	231	316	762	3827	19.9%	8.3%
Lung	674	260	326	850	4435	19.2%	6.6%
Pancreas	676	282	295	885	5463	16.2%	3.8%
Testis	731	319	339	962	4560	21.1%	7.7%
Spinal cord	721	300	359	947	5235	18.1%	5.6%
Thymus	866	366	447	1138	4865	23.4%	7.3%
Thyroid	415	99	201	474	2629	18.0%	8.1%
Adrenal gland	721	290	357	923	4862	19.0%	6.1%
BAT	603	249	297	780	4344	18.0%	6.2%
Blood	455	199	175	596	4389	13.6%	2.0%
Seminal vesicle	557	245	228	730	4542	16.1%	3.5%
Kidney	731	286	377	928	4629	20.0%	7.7%
Skeletal muscle	346	63	173	380	1704	22.3%	8.2%
Spleen	761	303	346	980	4814	20.4%	6.0%
Colon	697	320	305	918	5620	16.3%	4.2%
Skin	754	279	390	960	4927	19.5%	6.5%
Small intestine	692	263	340	868	3779	23.0%	10.6%
Heart	539	174	272	650	3179	20.4%	8.8%
Bladder	808	339	396	1056	6030	17.5%	6.2%
Stomach	650	264	311	832	5099	16.3%	4.7%
Tongue	426	107	212	488	2825	17.3%	5.1%
Liver Profiling	126	37	17	152	2175	7.0%	0.1%

* Total includes DNA binding protein, TC and TF; there were some overlap among them.

The ratio of abundance was the amount of TFs in total proteins.

We have added the TC and DBP quantitative identification and analysis in the revision, and also updated them in the TF Atlas website.

Q4: Line 130: DBTF is not defined.

Reply: DBTF is the abbreviation for DNA binding transcription factor. We have annotated this in the revision. To eliminate of ambiguity, we have added an abbreviation form in the revision (**Table CL7**). We have added the abbreviation index in the revision.

Table CL7 Abbreviation index

Abbreviation	Full name
TF	Transcription factor
DBTF	DNA binding transcription factor
ttr TFs	Tissue type restricted TFs
ttm TFs	Tissue type maintenance TFs
FOT	Fraction of total
NR	Nuclear receptor
ubiTFs	Ubiquitous TFs
non-ubiTFs	Non-ubiquitous TFs
TG	Target gene
PHx	Partial hepatectomy

Q5: In line 142: FOT is not defined.

Reply: FOT is the abbreviation of fraction of total. FOT is defined as a TF's iBAQ divided by the total iBAQ of all identified proteins in a particular tissue. Its definition is included in **Table CL7**.

Q6: In line the 155 the authors mention nuclear receptors (NRs) but in line 133 they talk about NHRs. Consistency should be kept throughout the manuscript.

Reply: Thanks for the comment. To keep the consistency, we uniformly use “nuclear receptor (NR)” in the revision.

Q7: What is the difference between ttrTFs and ubiquitous-non-uniform TFs? Some of the definitions are confusing and there are many acronyms in the paper making it hard to read.

Reply: We apologize for the confusion. In the revision, we have explained the terms used in this study, as follows:

TtrTFs: tissue type restricted TFs; TFs that are expressed in a particular tissue at levels that are at least 10 times higher than the median value of all adult tissues.

Ubiquitous TFs: TFs with a transformed median expression value of > 0.5 were considered ubiquitous TFs.

Ubiquitous-non-uniform TFs: Among ubiquitous TFs, the expression of twenty-seven TFs exhibited a maximum value of less than 10 times the median value, indicating a ubiquitous-uniform distribution.

Indeed, there is a considerable overlap between these two categories. However, these two methods classified TF patterns from a different perspective.

We have updated this item explanation of the TF sub-groups in the Methods section.

Q8: Some figures lack appropriate labels, and larger fonts would benefit reading. Figure 2f: MaxValue and MedianValue of what? Figure 3c needs a label in the y-axis. Figure 4a, 4d, 6c, 6e, 6f, 7c need a label for the color gradients. What are the axis in figure 5a? Figure 5e: label missing in top graph. Figure 6f: what is it being clustered? A label is missing in the y-axis of figure 7g.

Reply: We apologize for not labeling the figures clearly.

Figure 2f: We plotted the median and max value of TF DNA binding activates, transformed to

log₁₀ scale.

Figure 3c: We added the label “Expression FOT(log₁₀)” for the y-axis.

Figure 5a: The axis was the TFs which expressed in more than four tissues in 24 adult tissues.

Figure 5e: We added the annotation “Correlation coefficient” in the figure legend.

Figure 6f: These were top 30 differentially expressed TFs for each system.

Figure 7g: The y-axis represented the percentage of TFs with DMRs.

We have updated these figures and made the correct labels in the revision.

Q9: In line 223 the authors say they detected 47 NRs from 32 tissues. But in the following sentence they talk about half of adult tissues (24 in total). This is confusing.

Reply: In the “Transcription Network of the NRs” section, we investigated the NR DNA binding activities throughout the all tissues that we measured. We have removed the inaccurate words “in the adult animal”.

Q10: In paragraph 277-284 the authors use cosine similarity. The way it is defined is not very intuitive and it doesn’t scale linearly with the overlap in the set of tissues shared by two TFs. The authors should explore other more intuitive measures of similarity such as the Jaccard index or PCC.

Reply: Thanks for the comments. In our previous study, we used cosine similarity to evaluate protein relationships in core complex across selected IP-MS experiments (*Proc Natl Acad Sci U S A*, 2010, PMID: 20133760). So we used the same method to describe TF co-expression patterns. As suggested by the reviewer, we have converted the cosine similarity (angle) to Person correlation coefficient to make it easier to understand.

Q11: In paragraph 312-325 the authors mention the correlation coefficient for the expression of TF pairs. Is this based on catTFRE or mRNA expression? Why do the authors use PCC in this case and cosine for figure 5a?

Reply: It was based on catTFRE data. We have uniformly used PCC throughout the manuscript in revision.

Q12: Line 352-354: The correlation coefficient and the p-value should be included.

Reply: We have added the correlation and coefficient as suggested (Correlation coefficient was 0.9802, P = 0.0033, **Fig. 5C**)”.

Q13: Line 382: TG is not defined.

Reply: The target genes (TG) of the TFs are derived from a previously published paper CellNet (*Cell*, 2014, PMID: 25126793). We have added the details in the revision.

References:

1. Ding C, Chan D W, Liu W, et al. Proteome-wide profiling of activated transcription factors with a concatenated tandem array of transcription factor response elements [J]. *Proceedings of the National Academy of Sciences*, 2013, 110(17): 6771-6776.
2. Shi W, Li K, Song L, et al. Transcription Factor Response Elements on Tip: A Sensitive Approach for Large-Scale Endogenous Transcription Factor Quantitative Identification [J]. *Analytical Chemistry*, 2016, 88(24): 11990-11994.
3. Ravasi T, Suzuki H, Cannistraci C V, et al. An atlas of combinatorial transcriptional regulation in mouse and man [J]. *Cell*, 2010, 140(5): 744-752.
4. Uhlén M, Fagerberg L, Hallström B M, et al. Tissue-based map of the human proteome [J]. *Science*, 2015, 347(6220): 1260419.
5. Kim M S, Pinto S M, Getnet D, et al. A draft map of the human proteome [J]. *Nature*, 2014, 509(7502): 575-581.
6. Cahan P, Li H, Morris S A, et al. CellNet: network biology applied to stem cell engineering [J]. *Cell*, 2014, 158(4): 903-915.

7. Malovannaya A, Li Y, Bulyanko Y, et al. Streamlined analysis schema for high-throughput identification of endogenous protein complexes [J]. Proceedings of the National Academy of Sciences, 2010, 107(6): 2431-2436.

REVIEWERS' COMMENTS:

Reviewer #1 (Remarks to the Author):

The authors have done an excellent job responding to the raised concerns. By addressing these, they have significantly improved the manuscripts by clarifying some of their statements, describing some of the limitations of the used methodology, and including more data and comparative analyses to published data. Collectively, this is a comprehensive highly valuable data set that will be of great use for the large research community investigating transcriptional regulation. Therefore, I recommend acceptance of this manuscript for publication.

Reviewer #2 (Remarks to the Author):

In the revised manuscript by Zhou et al, the authors have addressed most of the concerns raised by reviewers #1 and #2. Overall this dataset will be of great use to the gene regulation scientific community. Some additional comments that would further improve the manuscript:

- 1) Although the authors added a paragraph in the Discussion about some of the limitations of the assay, a more indepth discussion needs to be included about how these limitations may or may not impact the results and conclusions obtained throughout the paper.
- 2) The full catTFRE sequence must be provided. This is especially important given that motifs for some TFs may be overrepresented while motifs may be missing for others.
- 3) The www.tfatlas.org URL is not functioning. This needs to be fixed.
- 4) Lines 222-225: No evidence is provided for the correlation between TF mRNA levels and DNA-binding activity.
- 5) Lines 401-403: Statement is unclear.
- 6) Lines 409-411: Overstatement

Reviewer #1 (Remarks to the Author):

The authors have done an excellent job responding to the raised concerns. By addressing these, they have significantly improved the manuscripts by clarifying some of their statements, describing some of the limitations of the used methodology, and including more data and comparative analyses to published data. Collectively, this is a comprehensive highly valuable data set that will be of great use for the large research community investigating transcriptional regulation. Therefore, I recommend acceptance of this manuscript for publication.

Reply: Many thanks for reviewer's positive comments.

Reviewer #2 (Remarks to the Author):

In the revised manuscript by Zhou et al, the authors have addressed most of the concerns raised by reviewers #1 and #2. Overall this dataset will be of great use to the gene regulation scientific community. Some additional comments that would further improve the manuscript:

1) Although the authors added a paragraph in the Discussion about some of the limitations of the assay, a more indepth discussion needs to be included about how these limitations may or may not impact the results and conclusions obtained throughout the paper.

Reply: We have added the more in-depth discussion about the limitations in the discussion section.

In the revision: "As an in vitro approach and being naked DNA template, the catTFRE approach does have its limitations. As naked DNA was used to measure the potential DNA binding activities of TFs, the response elements in the cell may be blocked in a nucleosome context or by other histone/DNA modifications. The "activity" of a TF as measured by catTFRE may not reflect its

actual activity in all loci on the chromosome. While the multiple TF response elements on catTFRE allows for the enrichment and identification of many TFs, this approach cannot distinguish the individual TF transcriptional machinery nor dissect the TC-TF complex, prohibiting the construction of the regulatory network between TFs and TCs. A more precise approach, for example, using single TF response element to purify TF-TC complexes may be applied to further validate functional modules drawn in this study.”

2) The full catTFRE sequence must be provided. This is especially important given that motifs for some TFs may be overrepresented while motifs may be missing for others.

Reply: We have added the full catTFRE sequence to the Supplementary Data 1 in the revision.

3) The www.tfatlas.org URL is not functioning. This needs to be fixed.

Reply: We have restored the website. The website works well now.

4) Lines 222-225: No evidence is provided for the correlation between TF mRNA levels and DNA-binding activity.

Reply: In the revision, we deleted the overstatements or not-so-accurate statements to ensure the integrity of the study.

5) Lines 401-403: Statement is unclear.

Reply: We removed these indistinct statements in the revision.

6) Lines 409-411: Overstatement

Reply: We removed these overstatements in the revision.